

# Role of the forcing sources in morphodynamic modelling of an embayed beach

Nil Carrion-Bertran[1], Albert Falqués[1], Francesca Ribas[1], Daniel Calvete[1], Rinse de Swart[1,*],
Ruth Durán[2], Candela Marco-Peretó[2], Marta Marcos[3], Angel Amores[3], Tim Toomey[3],
Àngels Fernández-Mora[4], and Jorge Guillén[2]

[1]Department of Physics, Universitat Politècnica de Catalunya, C/Jordi Girona, 1-3, 08034, Barcelona, Spain
[2]Institut de Ciències del Mar-CSIC. Passeig Marítim de la Barceloneta, 37, 08003 Barcelona, Spain
[3]Institut Mediterrani d'Estudis Avançats (UIB-CSIC). C/Miquel Marquès 21, 07190 Esporles, Illes Balears, Spain
[4]Balearic Islands Coastal Observing and Forecasting System (SOCIB). Parc Bit, Naorte, Bloc A, 07121 Palma de Mallorca,
Illes Balears, Spain
[*]now at WaterProof Marine Consultancy & Services BV, 8221 RC Lelystad, The Netherlands

**Correspondence:** Nil Carrion Bertran (nil.carrion@upc.edu)

**Abstract.** The sensitivity of a 2DH coastal area (XBeach) and a reduced-complexity (Q2Dmorfo) morphodynamic models to using different forcing sources is studied. The models are tested by simulating the morphodynamic response of an embayed beach in the NW Mediterranean over a 6-month period. Wave and sea level forcing from in-situ data, propagated buoy measurements, hindcasts as well as combinations of these different data sources are used and the outputs are compared to in-situ
bathymetric measurements. Results show that when the two models are calibrated with in-situ measurements, they accurately reproduce the morphodynamic evolution with a "Good" BSS (Brier Skill Score). The wave data propagated from the buoy also produces reliable morphodynamic simulations but with a slight decrease in BSS. Conversely, when the models are forced with hindcast wave data the mismatch between the modelled and observed beach evolution increases. This is attributed to a large extent to biased mean directions in hindcast waves. Interestingly, in this small tide site the accuracy of the simulations did not depend on the sea-level data source, and using filtered or non-filtered tides also yielded similar results. These results have implications for long-term morphodynamic studies, like those needed to validate models for climate change projections, emphasizing the need of using accurate forcing sources such as those obtained by propagating buoy data.

## 1 Introduction

Coastal zones, the boundary between ocean and land, are one of the most dynamic geological systems in our planet (Neumann et al., 2015). Their enormous socio-economic and ecological values have always attracted human settlements and development, which is why coastal areas are the most populated regions in the world (Martínez et al., 2007). This is specially true in the Mediterranean basin (Lionello et al., 2006). However, the intensification of human interests and activities in these areas have also increased the amount of infrastructures, which often incremented the vulnerability of the coastal areas to flooding and erosion processes (Adger et al., 2005). Sea-level rise is expected to produce an increment of inundation events and aggravate the erosion trends, especially in low-lying sandy beaches (Vousdoukas et al., 2016; Ranasinghe, 2016; Oppenheimer et al., 2019).



Consequently, understanding the sandy beach responses to climate change has become a critical issue in the future coastal management context (Nicholls et al., 2016; Hinkel et al., 2018). In particular, forecasting such climate change impacts during the forthcoming decades and beyond is a major scientific challenge that will strongly benefit from reliable morphodynamic predictions.

There are different methods for assessing long-term beach evolution with various degrees of accuracy (Montaño et al., 2020). These range from fully data-driven to fully physically-based models (Luijendijk et al., 2017). A common approach is using morphodynamic models, and among them, the most appropriate one must be selected to simulate the physical processes with the desired accuracy (Ranasinghe, 2020). The simplest option is the Bruun Rule (Bruun, 1962), although it should be used with caution because it ignores many important processes such as the gradients in longshore transport and the short-term
climate variability (Cooper and Pilkey, 2004; Ranasinghe et al., 2012; Luque et al., 2023). Coastline models (Robinet et al., 2018), which solve the morphodynamics with simplifications by describing only a few dominant processes, are suitable for long-term simulation although their skills are also limited (Montaño et al., 2020). 2DH coastal area models, such as XBeach (Roelvink et al., 2009), resolve the relevant hydrodynamic and morphodynamic processes within the surf and shoaling zones and successfully describe the physical mechanisms that govern the beach systems at the desired space scale (Kombiadou
et al., 2021). However, they require much higher computational capacity than coastline models, making them unsuitable for long-term simulations (Karunarathna and Reeve, 2013). In between coastline and 2DH coastal area models, there are reduced-complexity models, such as Q2Dmorfo (van den Berg et al., 2011; Arriaga et al., 2017) which is designed to simulate the shoreline evolution at large spatial and temporal scales. It computes wave transformation and topobathymetric evolution with the important simplification that surf zone hydrodynamics are not resolved, the sediment fluxes are computed parametrically
from the wave field. The advantage is that the computational cost is significantly reduced with respect to 2DH models while maintaining a reasonable accuracy (Ribas et al., 2023). For all morphodynamic models, an initial morphology of the beach and the external wave conditions and sea level forcing, as well as the calibration and validation of the model itself, are required.

Ideally, the model forcing should be based on data from in-situ instruments. However, these data are not always available at the desired location and may not cover all the required time period. Alternatively, wave data can be obtained by propagating
buoy measurements or by using data from global hindcast models. Often, a combination of different data sources is used as forcing. In the case of future projections under climate change scenarios external forcing conditions are generated from large datasets with the corresponding uncertainty associated to different forcing realizations (Angnuureng et al., 2017; Antolínez et al., 2018). Despite the importance and variety of forcing sources, to our knowledge, the morphodynamic effect of using different forcing sources has not yet been studied. Additionally, the sensitivity to using various sources can differ among the
models used. As 2DH models predict the beach dynamics in more detail, they could be more sensitive when an inaccurate external forcing source is applied, resulting in a poorer outcome. In contrast, a reduced-complexity model may be less affected by inaccuracies in the wave or sea-level inputs, as it filters out small scale processes that, if inaccurately described, could spoil the large scale behaviour. Therefore, a central question is how the different forcing sources affect different types of morphodynamic models.





The assessment of long-term climate change impacts on beaches has to be performed at local to regional scales and on specific types of beaches (Ranasinghe, 2020; Sánchez-Artús et al., 2023). In the Catalan coast (Northwestern Mediterranean Sea), beaches are often embayed by natural or anthropogenic structures (e.g., headlands or groins, respectively), limiting or avoiding the sediment transfer to/from the nearby littoral cells. These structures also provide protection to wave action, making obliquely incident waves that reach the shore less energetic. Thus, embayed beaches should be less vulnerable to oblique

storm impacts in comparison to the non-protected open beaches. On the other hand, the fact that they do not receive external sediment supply can worsen their vulnerability to sea level rise (Monioudi et al., 2017). But in general, the adaptation of sheltered beaches to different climatic conditions that include global warming scenarios with higher sea levels has been barely investigated (Toimil et al., 2020).

The aim of this study is to quantify the effect of using different sources for the forcing conditions in morphodynamic

modelling of an embayed beach at time scales of several months. This will be approached by applying the 2DH XBeach model and the reduced-complexity Q2Dmorfo model to a Mediterranean embayed beach during a 6-month period. This time period is an intermediate duration between the short term (adequate for XBeach) and the long term (adequate for Q2Dmorfo), meaning that for this duration the range of both models roughly meet. The manuscript is organized as follows: Sect. 2 describes the available in situ wave and sea level data sets and the two topo-bathymetric surveys conducted in Castell beach, Palamós (NW

Mediterranean Sea, Catalunya, Spain). Then, the models used, the chosen setup and the calibration method performed using the in situ source are presented (Sect. 3). In Sect. 4 the outcomes of the calibration of the two models are shown and in Sect. 5 the sensitivity of the two models to using different forcing sources is presented. Section 6 includes a discussion, with a comparison between the two models and with previous studies, and the conclusions are listed in sect. 7.

## 2   Study site and data

### 2.1   Site description

This study focuses on Castell beach, a sandy embayed beach located next to Palamós, at the Catalan Costa Brava in the north western Mediterranean Sea (Fig. 1). The beach shore normal is roughly oriented towards south (at $190°$ from north). The dry beach is $\sim 300$ m long and $\sim 80$ m wide, and a median grain size of $d_{50} = 0.4$ mm is representative of the submerged active zone. It is bounded by two rocky headlands that extend $\sim 100$ m and 160 m from the shoreline on its west and east sides

respectively. The small Aubi creek reaches Castell beach from the north. It is usually dry, but during episodes of heavy rain it can transport water and sediment to the coast, changing its morphology.

The Catalan coast is an area of low to intermediate wave energy, where calm periods are dominant during most of the year, especially during spring and summer. Storms, which are usually observed during autumn and winter, are here defined as periods of more than 12 h with significant wave height ($H_s$) exceeding 1.5 m and a $H_s$ peak exceeding 2.5 m in deep water (Ojeda

and Guillén, 2008). The highest energy events usually reach the Catalan coast from the east coinciding with the direction of the maximum available fetch (Sánchez-Arcilla et al., 2008). Only southerly and easterly waves can reach Castell beach due to the geometry of the surrounding rocky headlands, and the latter must undergo substantial refraction to arrive at the beach. The



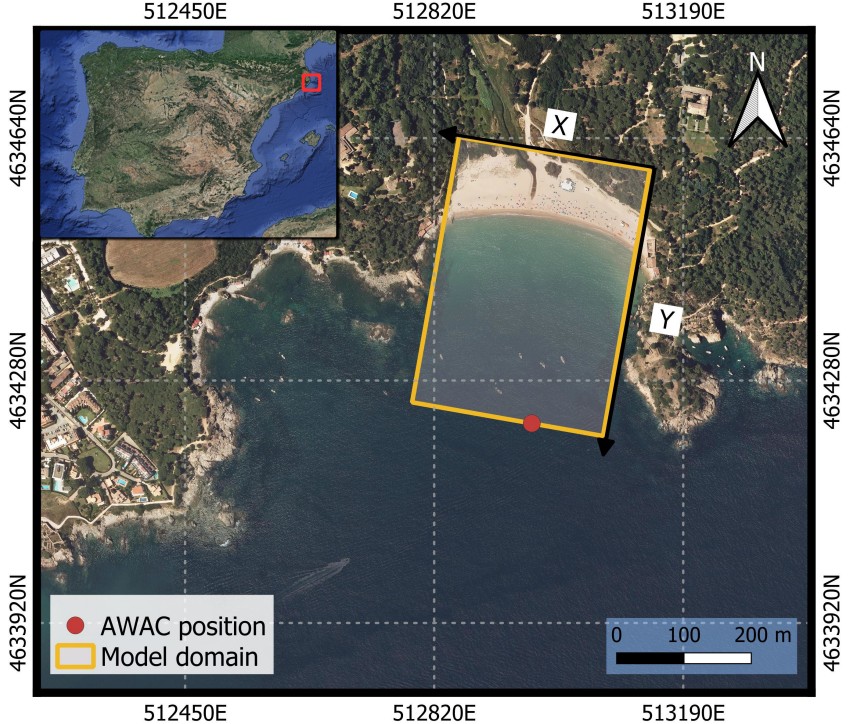

**Figure 1.** Map of the study site showing the domain of the morphodynamic models and the AWAC position. Arrows show the local coordinate system $(x, y)$ used in this study. (Source: © Google Earth, Image from Institut Cartogràfic i Geològic de Catalunya).

astronomical tidal range in the Catalan coast is $\sim 20$ cm (Simarro et al., 2015) while meteorological tides (storm surges) can reach $\sim 40$ cm (return period of 1 year, Toomey et al. (2022)).

## 2.2 Topo-bathymetric data

Two topo-bathymetric surveys were conducted on January 28th and July 8th 2020 (Fig. 2). Bathymetry was measured with a Hypack ® multibeam echo-sounder and a GNSS antenna mounted on a 6m LOA pneumatic boat, covering the beach embayment extent from approximately 1 m to 20 m depth. Echo-sounder measurements were processed using Hypack ® software. An initial automatic filter was applied to eliminate any spike outliers. Adjustments for head, pitch, roll, and heave were au-
tomatically applied. A human-eye review of the echo-sounding measurements was also conducted to remove noise sounding. RTK-GPS topo-bathymetric measurements were added to the sounding points cloud for a second review of data to check elevation matching of the common points between RTK-GPS and the echo-sounder. The full data set was then extracted considering cell points of 1x1 m in the post-processed 3D point cloud files. All the topobathymetric data were referred to the Geoid EGM08D595, from the "Institut Cartogràfic i Geològic de Catalunya".





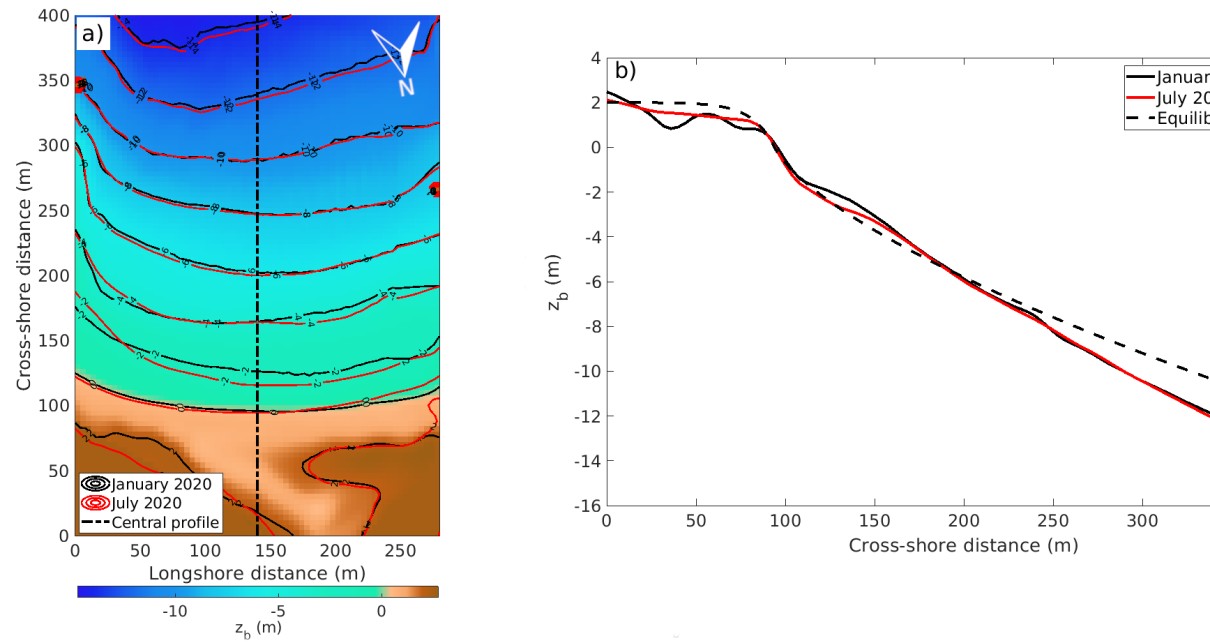

**Figure 2.** Topo-bathymetric surveys in January and July 2020 within the model domain oriented using the local coordinate system $(x, y)$ shown in Fig. 1 (panel a), and their central cross-shore bathymetric profiles (panel b). The dashed line in the right panel displays the equilibrium profile used in the Q2Dmorfo model. In both panels January is represented in black lines and July in red lines. The background colours in the left panel correspond to January 2020.

The two measured topo-bathymetries differed mainly in the shallower area up to $4$ m depth, the latest showing a certain overall retreat of the nearshore and shoreline anticlockwise rotation. The slope of the swash zone was of approximately $\beta_s = 0.16$ (Fig. 2b) and, at greater depths, the slope decreased to approximately $0.05$. The berm reached a height of about $2$ m and the dry beach displays the footprint of the creek channel. Most of the observed changes in the dry beach were probably related to the creek position modifications during the 6 months between the two topo-bathymetries (see the Supplementary Information).

Notice that the 2 months before the first survey were highly energetic, ending with the Gloria storm on 19-26 January 2020 (Amores et al., 2020; Sancho-García et al., 2021; Pérez-Gómez et al., 2021), the strongest storm in at least 30 yr that affected the Mediterranean beaches of Spain coming from the northeast with significant wave heights up to $8$ m.

### 2.3    Wave data

During the 6-month time lapse between the two topo-bathymetric surveys, hourly wave and sea-level data were measured by
a Nortek® Acoustic Wave and Current Profiles (hereinafter, AWAC) deployed at 14.5 m depth (red circle in Fig. 1). This equipment combines a bottom-mounted upward-facing Doppler current profiler (ADCP) with a directional wave gauge. The ADCP measures directional currents along the water column, while directional wave parameters are computed using pressure time-series, acoustic surface tracking (AST) and surface velocity. The frequency spectrum and other non-directional wave





parameters are estimated using these measurements (Pedersen et al., 2007; de Swart et al., 2020). The wave measurement

setup used 1200 samples at 1 Hz starting at the beginning of each hour. Raw data was processed by Nortek QuickWave® software which provided the main wave parameters (non-directional and directional spectrum), surface currents and mean sea-level (Fig. 3).

To test the sensitivity of the forcing sources, two other wave sources were used. The first one was obtained by propagating wave conditions measured by the Cap Begur wave buoy (located at 3.65°E 41.9°N at a water depth of 1200 m) to the AWAC

location (at 14.5 m depth) using the SWAN wave model version 41.31 (Booij et al., 1999, SWAN Team, 2019), following a methodology similar to that of De Swart et al. (2021) (see the Supplementary Information for the details on the methodology). The second additional wave data was obtained from the CoExMed hindcast, generated using the fully-coupled hydrodynamic-wave model SCHISM (Zhang et al., 2016) forced by the atmospheric pressure and surface wind from ERA5 (Hersbach et al., 2020) over the Mediterranean Sea (Toomey et al., 2022). The CoExMed data set consists of hourly wave bulk parameters,

significant wave height $H_s$, peak period $T_p$ and wave peak direction $\theta_p$ spanning the period 1950-2021 with a spatial resolution down to 200 m in coastal areas. Notice that the CoExMed wave direction is the peak direction. Nevertheless, the wave peak and mean directions were compared and there were not significant differences. Thus, from now on, the wave peak direction from CoExMed is going to be referred to as mean direction in concordance with the two other wave forcing sources. Here, a specific SCHISM simulation was performed to obtain the data at the location of the AWAC. The averaged wave characteristics

of the three sources are shown in Table 1.

The 6 months of the study were generally not very energetic but some episodes of medium wave intensity occurred (Fig. 3). In early March, a storm reached the coast from about 160° N with a maximum $H_s$ of 2.5 m, the highest value recorded during the measured period. In fact, waves arrived from the S and SSE a significant percentage (55 %) of the studied time, which is not particularly common on the Catalan coast, where eastern direction tends to dominate, but Castell beach orientation favours

the entrance of the southern directions. From mid-March to mid-April, several low-energetic storms reached the coast from the E (turning to SE in front of the beach due to refraction) with $H_s$ above 1.5 m. In mid-May and mid-June, two low-energy storms (maximum $H_s$ of 1.5 m) reached the coast from the SSE, but these last two months were generally characterized by low-energetic wave conditions.

The three wave data sources provide similar values for the significant wave height (Fig. 3a). The peak period obtained from

the propagation of the Cap Begur buoy data using SWAN overestimates the in situ values whereas the data obtained from CoExMed underestimates them by a similar amount of about 0.5 s (Table 1). The mean directions are better represented by the propagation of buoy data using SWAN than by the CoExMed hindcast. The latter (former) overestimates the angles from the southern waves with a bias of 18° (3°) to the south-southwest and a root mean square error ($\varepsilon$) of 35° (20°) (Fig. 3c and Table 1).

**2.4 Sea level data**

Three sea level data sets were used. The first one was measured in situ by the AWAC, the second one was obtained from the Barcelona (BCN) harbour tide gauge (a radar Miros sensor managed by "Puertos del Estado" from the Spanish Government),



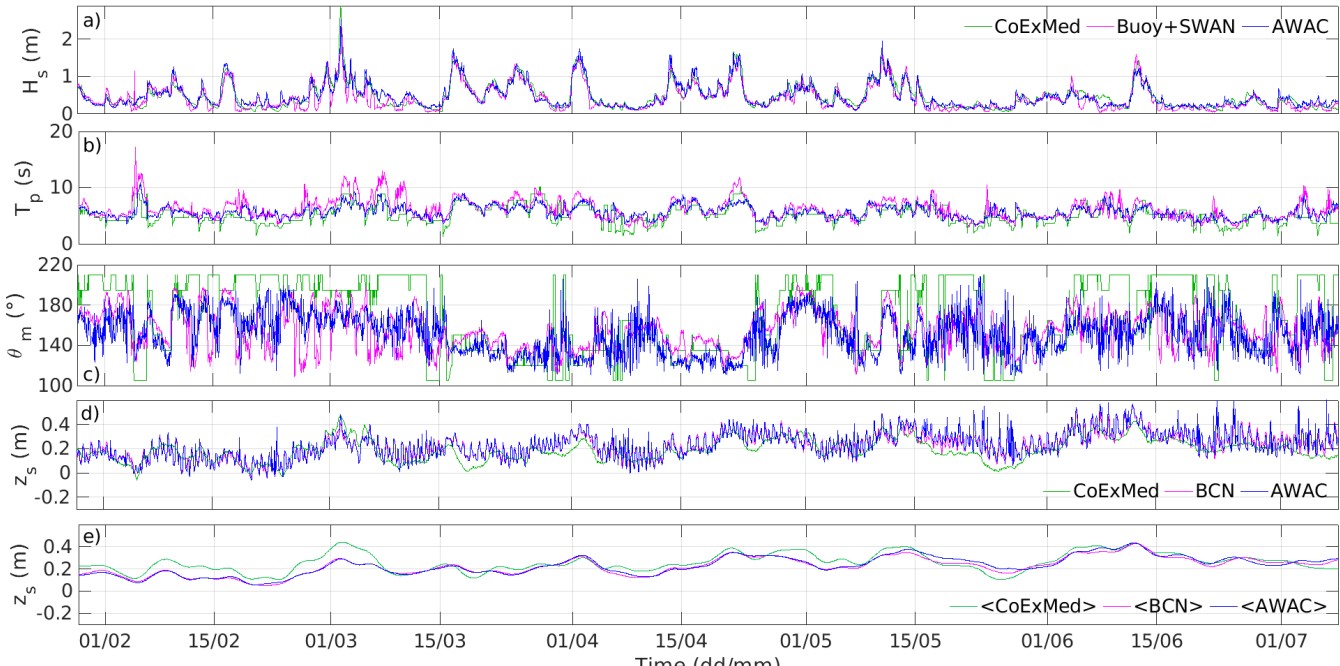

**Figure 3.** Data at the AWAC location of the different forcing sources during the 6-month study period. Time series of significant wave height ($H_s$, panel a), peak period ($T_p$, panel b), and mean wave direction with respect to north ($\theta_m$, panel c) are shown for the three wave data sources. Time series of instantaneous sea level data from the AWAC, the Barcelona harbour tide gauge and CoExMed hindcast are shown (panel d), as well as the 5-day averaged sea level data from these two latter sources together with instantaneous sea level data obtained from CoExMed, also shown in the previous panel (panel e).

which is located ∼ 100 km far from the study area, and the third one was extracted from CoExMed data set (Fig. 3d-e). Sea-level from AWAC pressure time-series was computed assuming hydrostatic conditions above the instrument, constant water

temperature and density along the water column, and considering the depth of the instrument deployment and the height from the sea-bed of the pressure sensor (65 cm). Apart from the wave conditions, the 72 yr hindcast by CoExMed also generated a sea level time series described above. This was done by using the effects of mean sea level atmospheric pressure, surface winds, waves on total sea surface elevation and not including the astronomical tide frequencies, for the period 1950-2021 and with hourly temporal sampling. Finally, a 5-day running average in the sea level time series of the three sources was performed

in order to test the role of the high-frequency (mostly controlled by tides as defined here) sea level variability (Fig. 3e). All sea level data were referred to the same Geoid as the topobathymetric data, in order to have all the model inputs referred to the same datum.

The AWAC instrument sank 0.5 metres from the initial position where it was deployed during a storm in early March. This affected the sea level measurements causing an upward bias in the data recorded since then. In order to fix this problem,

the AWAC sea level data was adjusted to reproduce the monthly trends of the Barcelona harbour tide gauge. First, the two





**Table 1.** Wave characteristics of the different data sources at the AWAC location, $\overline{H}_s$ being the mean significant wave height, $H_{s,max}$ the maximum significant wave height, $\overline{T}_p$ the mean peak period and $\overline{\theta}_m$ being mean wave direction with respect to north. The root mean square error ($\varepsilon$) of the propagated buoy and CoExMed data compared to the AWAC data is also included.

| Wave data source | $\overline{H}_s$ (m) | $H_{s,max}$ (m) | $\varepsilon_H$ (m) | $\overline{T}_p$ (s) | $\varepsilon_T$ (s) | $\overline{\theta}_m$ (°) | $\varepsilon_\theta$ (°) |
|---|---|---|---|---|---|---|---|
| AWAC | 0.48 | 2.52 | - | 5.7 | - | 151 | - |
| Buoy + SWAN | 0.47 | 2.47 | 0.14 | 6.1 | 1.2 | 154 | 19.9 |
| CoExMed | 0.42 | 2.87 | 0.13 | 5.1 | 1.2 | 169 | 34.8 |

time series were smoothed (to focus on the monthly trends) and subtracted. Then, a hyperbolic function was adjusted to the differences to finally subtract this function to the original AWAC data.

## 3 Description of the morphodynamic models

### 3.1 XBeach equations

XBeach is an open-source 2DH morphodynamic model initially designed to simulate the storm impact on dunes and barrier islands (Roelvink et al., 2009), although it is nowadays being applied to describe multiple coastal processes. The model determines wave transformation over the evolving bathymetry and solves the mean surf-zone hydrodynamics. It then computes the associated sediment transport and the induced seabed evolution. It is suited to model beach morphodynamics at relatively short time scales of days-weeks. A brief description of the equations and parameterisations used within XBeach, especially focusing

on sediment transport and bed evolution, is presented in the following paragraphs and a full description of the model can be found in the literature (e.g., van Thiel de Vries, 2009; de Vet, 2014; Elsayed and Oumeraci, 2017).

The model propagates the short waves using the time-dependent wave-action balance equation and the roller equation. In these equations, the directional distribution of the wave-action density is taken into account, whereas the frequency spectrum is characterized by a single representative value. Three wave modes are implemented in XBeach. The `stationary` one resolves

the wave-averaged equations, without including the infragravity waves associated to the short wave action. There is also a non-stationary mode called `surfbeat`, that simulates the short wave variations on the wave group scale and their associated long waves. The third mode is the `non-hydrostatic` one, which resolves individual waves, but it was discarded in this study due to its high computational cost.

The low-frequency currents and sea surface level are determined using the nonlinear shallow water momentum and mass

balance equations, using a Generalized Lagrangian Mean formulation and including all relevant forces (e.g., wind, waves, bed friction and turbulent diffusivity). The main dynamic variables are the water depth $D$ and the depth-averaged water velocity $\boldsymbol{v}^L$, which is called Lagrangian velocity in XBeach terminology. The model also uses a second velocity (called Eulerian in



their terminology), $\boldsymbol{v}^E = \boldsymbol{v}^L - \boldsymbol{v}^S$, which is the depth-averaged velocity minus the Stokes drift velocity $\boldsymbol{v}^S$, computed out of the wave and roller energies (van Thiel de Vries, 2009). Bed shear stresses are computed using the formulation by Ruessink et al. (2001), written as a function of the velocity $\boldsymbol{v}^E$ (for being more representative of the current near the bed) and the bottom friction coefficient, $c_f$. The latter is modelled using the depth-dependant Manning formulation, $c_f = g\,n^2/D^{1/3}$, where $g$ is gravity and $n$ is a coefficient that can be varied.

An advection-diffusion equation (Galappatti and Vreugdenhil, 1985) is solved to compute the depth-averaged sediment volumetric concentration $c$,

$$\frac{\partial(c\,D)}{\partial t} + \nabla \cdot \left( c\,D\,(\boldsymbol{v}^E + u^A \hat{k}) + \nu_h\,D\,\nabla c \right) = D\,\frac{c_{eq} - c}{T_s}\,, \tag{1}$$

Here, $u^A$ is a velocity magnitude representing the wave nonlinearity, $\hat{k}$ is the wave direction and $\nu_h$ is the horizontal eddy viscosity

that is used both here to represent a sediment diffusion coefficient and in the water momentum balance. The wave nonlinearity velocity is expressed as

$$u^A = (f_{Sk}\,S_k - f_{As}\,A_s)\,u_{rms}\,, \tag{2}$$

where $S_k$ and $A_s$ describe the skewness and asymmetry in wave motion, respectively (computed following van Thiel de Vries (2009)), $u_{rms}$ is the standard RMS wave orbital velocity near the bed and $f_{Sk}$ and $f_{As}$ are two important calibration parameters. Moreover, $c_{eq}$ in the RHS of Eq. (1) is the depth-averaged equilibrium sediment concentration and $T_s$ is an adaptation time (for the concentration to reach the equilibrium value) computed as a function of water depth and sediment fall velocity (van Thiel de Vries, 2009). Several formulations can be used for $c_{eq}$ and we chose the XBeach default one, the Van Thiel-Van Rijn equation, which reads

$$c_{eq} = \frac{A_{sb}}{D} \left( \sqrt{|\boldsymbol{v}^E|^2 + 0.64(u_{rms}^2 + 1.45k_b)} - u_{cr} \right)^{1.5} + \frac{A_{ss}}{D} \left( \sqrt{|\boldsymbol{v}^E|^2 + 0.64(u_{rms}^2 + 1.45k_b)} - u_{cr} \right)^{2.4}\,, \tag{3}$$

where $A_{sb}$ and $A_{ss}$ are the bed load and suspended load parameters (van Thiel de Vries, 2009) and $u_{cr}$ is the critical velocity, computed as a weighted summation of the separate contributions by currents and waves. The sediment is assumed to be stirred by currents, waves and turbulence, where $k_b$ is the near-bed turbulence energy. The latter is an important source of sediment resuspension under breaking waves (Ribas et al., 2011) and is modelled following Roelvink and Stive (1989).

Finally, the seabed evolution is computed by solving the Exner equation,

$$\frac{\partial z_b}{\partial t} + \frac{f_{mor}}{1-p}\,\nabla \cdot \boldsymbol{q} = 0\,, \tag{4}$$

where $z_b$ is the bed level, $f_{mor}$ is the morphological acceleration factor, $p = 0.4$ is the porosity, and $\boldsymbol{q}$ is the total volumetric flux (or transport) of sediment and reads

$$\boldsymbol{q} = c\,D\left( \boldsymbol{v}^E + u^A \hat{k} \right) + \nu_h\,D\,\nabla c - f_{sl}\,c\,D\,|\boldsymbol{v}^L|\,\nabla z_b. \tag{5}$$



The last term represents the bed slope effect with $f_{sl}$ being the corresponding parameter. Notice that equations (1), (3), (4) and (5) must be consistent with the conservation of sediment. This only occurs if $\frac{\partial(cD)}{\partial t} = 0$ but, since this term is typically small, the error committed is minor. Besides, a reference bed slope of the swash zone, $\beta_s$, can also be provided so that the

swash zone profile tends towards it where $H/D > 1$, when working in `surfbeat` mode. Finally, an avalanching algorithm is also used in XBeach to account for the sediment collapse occurring during storm-induced dune erosion (Roelvink et al., 2009).

### 3.2  XBeach setup

A rectangular domain was used, localized as shown in Fig. 1, with the cross-shore coordinate rotated $190°$ with respect to north to adequately represent the Castell beach area and rocky headlands. The grid had an alongshore extension of $280$ m

and a cross-shore extension of $400$ m. Several grid resolutions were initially tested and the optimum values were found to be 4x4 m. Smaller resolutions result in a too high computational cost and larger ones were not accurate enough to describe the shallower parts of the domain. The position where the AWAC was deployed corresponded with the domain offshore limit. Thereby, the wave and sea level conditions available at the AWAC location (Sect. 2) could be directly applied at the seaward side of the domain. The headlands were simulated in XBeach with 2x2 non-erodible cells located at the offshore end of each

headland (at $344$ m and $264$ m from the $x$-axis in the east and west, respectively). These cells can not be eroded for being like solid structures and the incident waves are influenced by them as they propagate from the offshore boundary to the coast. This definition represents properly the wave shadow effects due to the presence of rocky headlands. Additionally, the lateral boundary conditions on the model were set as no-flux conditions for water and sediment.

Preliminary tests showed the importance of including the effects of wave groupiness to model Castell beach bed evolution.

The `stationary` mode presented a systematic erosion in the surf zone inducing an unrealistic recession of the coastline (compared to the final measured bathymetry). The `surfbeat` mode, on the other hand, could simulate the beach response to the incoming waves with a more realistic onshore transport in the surf zone minimising the shoreline recession, in agreement with the literature (Rutten et al., 2021; Bae et al., 2022) and was thereby selected for this study. When this mode is used, XBeach generates random wave time series within the spectral wave boundary condition that include wave groupiness. Then,

waves entering the domain are slightly different for each particular simulation, even when running exactly with the same model setup, imitating the stochastic nature of a real sea. In fact, this only occurs if a XBeach parameter called `random` equals 1. This of course affects beach dynamics: since the incident waves slightly change in each "particular simulation", the sediment transport is also modified, and the beach response can be different with exactly the same model setup. These small changes can accumulate over time and become significant when a large period of time is simulated, like in the present

study. The effect of the `random` parameter was hardly evaluated in previous studies because either shorter time periods were simulated or this randomness was simply disabled (`random`$= 0$) and therefore the same wave time series was always applied. In Rutten et al. (2021) the `random` mode was enabled and they demonstrated the importance of including wave stochastic behaviour for the morphodynamic evolution of a beach. Using `random`$= 0$ also proved to be inaccurate in the present study, as it only reproduces a specific offshore wave condition that lead to a particular result, which might not be representative to

the real stochastic character of the waves propagating to the shore, and does not take into account other potential realizations.





**Table 2.** List with several of the parameter values used in the XBeach model.

| Parameter | Symbol | XBeach name | Def. value | Units | Range tested |
|---|---|---|---|---|---|
| Offshore long wave randomness | - | `random` | 1 (enabled) | - | - |
| Wave computation mode | - | `wavemode` | `surfbeat` | - | - |
| Bed friction coefficient | $n$ | `bedfriccoef` | 0.03 | $m^{-1/3}$ s | [0.02 - 0.04] |
| Morphological factor | $f_{mor}$ | `morfac` | 10 | - | [5 - 20] |
| Swash zone slope | $\beta_s$ | `bermslope` | 0.16 | - | - |
| Near-bed turbulent energy mode | $k_b$ | `turb` | `wave_averaged` | - | - |
| Wave skewness factor | $f_{Sk}$ | `facSk` | 0.55 | - | [0.30 - 0.60] |
| Wave asymmetry factor | $f_{As}$ | `facAs` | 0.35 | - | [0.20 - 0.50] |

To adequately deal with the effect of using the `surfbeat` mode, the randomness in the offshore wave groupiness had to be handled. In the present study we used `random`= 1, and a series of 15 to 30 realizations were made to account for the corresponding variability in the beach response.

Different values of the parameter $n$ (from 0.02 to 0.04 $m^{-1/3}$ s, Table 2) in the Manning formulation for the bottom friction
coefficient $c_f$, were tested. These are typical values used to simulate the bed friction of sandy beaches (e.g., Schambach et al., 2018; Passeri et al., 2018; Kombiadou et al., 2021). The preliminary assessment of this parameter showed that the best value to represent the bed friction in Castell beach is 0.03 $m^{-1/3}$ s.

Preliminary tests concluded that the effects of the turbulence induced by the wave breaking on the equilibrium sediment concentration, represented by the parameter $k_b$ in Eq. (3), had to be computed with the `wave_averaged` mode. Either using
the `bore_averaged` mode or switching off this parameter increased the unrealistic erosion overestimation of XBeach. The default value for the morphological acceleration factor $f_{mor}$ in Eq. (4) used in this study was 10, reducing the computational time of the model, and values of 5 and 20 were also tested. The value of the swash zone slope measured in the January 2020 topography was applied, $\beta_s = 0.16$. The default value of the bed slope parameter in Eq. (5) was used, $f_{sl} = 0.15$ (de Vet, 2014).

Setting appropriate values (higher than the default one, i.e., 0.01) of the parameters $f_{Sk}$ and $f_{As}$ in Eq. (2) is essential
to increase onshore sediment transport and reproduce the post-storm beach recovery, in order to prevent the overestimation of erosion by XBeach model (Schambach et al., 2018; Kombiadou et al., 2021). In the present study, for the $f_{Sk}$ parameter, values between 0.30 and 0.60 with an increment of 0.05 were applied during the calibration, and the range of values used was between 0.20 and 0.50, also with an increment of 0.05, for $f_{As}$.

## 3.3 Q2Dmorfo equations

Q2Dmorfo is a reduced-complexity coastal morphodynamic model especially designed for large spatio-temporal scales (up to tens of km and decades). Its essential simplification with respect to 2DH models (e.g., XBeach) is that the mean hydrodynamics is not resolved, so that the sediment fluxes are computed parametrically from the wave field. On the other hand, in contrast with



one-line coastline models, the full topo-bathymetry is handled by solving the sediment conservation Eq. (4). Its most important equations are described in this section and a full description can be found in Arriaga et al. (2017). A Cartesian coordinate

system is used, with the $x$-axis pointing alongshore, the $y$-axis pointing seaward and the $z$-axis pointing upward. Notice that the coordinate axes $x - y$ are here rotated with respect to the common model description (see, e.g., Arriaga et al. (2017)). The sea bed is located at $z = z_b(x, y, t)$ and the mean sea level is at $z = z_s(x, y, t)$.

The model solves Eq. (4), with $p = 0.4$ and $f_{mor} = 1$, to compute the evolution of the bed level. The total volumetric flux of sediment $\boldsymbol{q}$ is assumed to be composed of longshore $\boldsymbol{q_L}$, cross-shore $\boldsymbol{q_C}$ and diffusive $\boldsymbol{q_D}$ components,

$$\boldsymbol{q} = \boldsymbol{q_L} + \boldsymbol{q_C} + \boldsymbol{q_D} \ . \tag{6}$$

At each point, the local "cross-shore" direction is defined by a unit vector $\hat{\mathbf{n}}$ perpendicular to a local smoothed bathymetric contour and directed offshore (see Arriaga et al. (2017) for details), and the local mean "alongshore" direction $\hat{\mathbf{t}}$ is defined so that the local system is orthonormal and right-handed.

The first term in Eq. (6) is the sediment transport related with the wave-induced longshore current and it is based on the

CERC formula (Komar, 1998)

$$\boldsymbol{q_L} = \mu H_b^{5/2} \left( \sin(2\alpha_b) - \frac{2r}{\beta_c} \cos(\alpha_b) \frac{\partial H_b}{\partial x} \right) f(y') \hat{\mathbf{t}} \ , \tag{7}$$

where $H_b$ is the RMS wave height at breaking, $\alpha_b = \theta_b - \phi_s$ is the angle between the wave direction at breaking and the local shore normal, and $\mu$ is a calibration parameter which is proportional to the standard CERC constant $K$ (Arriaga et al., 2017). The additional term proportional to the gradient of $H_b$ is relatively uncommon but has been here included to account for

the alongshore gradients in wave setup and is controlled by the $r$ parameter (Horikawa, 1988). Finally, $f(y')$ is a normalized cross-shore shape function, assumed to mimic the longshore current profile. Here, $y'$ is the distance from the closest coastline location to the point and $\beta_c$ is the actual beach slope at the shoreline. The second term in Eq. (6) parameterises the cross-shore transport by assuming a bathymetric tendency to evolve to a prescribed alongshore-uniform equilibrium profile, with $\boldsymbol{q_C}$ being proportional to the difference between the equilibrium slope $\beta_e$ and the actual local slope in the local cross-shore direction,

$$\boldsymbol{q_C} = -\gamma(\nabla z_b \cdot \hat{\mathbf{n}} + \beta_e)\hat{\mathbf{n}} \ . \tag{8}$$

The first term describes the downslope transport and the second term simulates the net wave-induced onshore transport (Falqués et al., 2021). The third term in Eq. (6) represents the tendency of small bumps to be flattened in the alongshore direction due to wave stirring if there is no positive feedback,

$$\boldsymbol{q_D} = -\gamma(\nabla z_b \cdot \hat{\mathbf{t}})\hat{\mathbf{t}} \ . \tag{9}$$

The stirring factor $\gamma$ in both $\boldsymbol{q_C}$ and $\boldsymbol{q_D}$ accounts for sediment stirring by currents, wave orbital velocity and turbulence. The magnitude of the horizontal momentum mixing given by Battjes (1975) is used as scaling factor,

$$\gamma = \nu \gamma_b^{-1/6} H_b^{11/6} Y_b'^{-1/3} g^{1/2} \psi(D) \ , \tag{10}$$



where $\gamma_b$ is the saturation ratio of $H/D$ inside the surf zone (here, $\gamma_b = 0.5$), $D = z_s - z_b$ is water depth, $Y_b'$ is the surf zone width (computed in the $y'$ direction), $g$ is gravity acceleration and the constant of proportionality $\nu$ is the second calibration

parameter. The shape function $\psi$ (Arriaga et al., 2017) is assumed to have a maximum value at the shoreline ($\psi(0) = 1$) and to decay both landward (across the swash zone) and seaward, being negligible at the depth of closure, $D_c$.

Incident monochromatic waves with $T = T_p$ (peak period), $H = H_s$ (significant wave height) and a wave angle $\theta$ are considered at the offshore boundary. Since sediment transport computation requires the wave characteristics at breaking, the waves are propagated inside the domain up to breaking point using the geometric optics approximation, i.e., applying the dispersion

relation, the wave number irrotationality and the wave energy conservation (van den Berg et al., 2012; Arriaga et al., 2017). From the computed wave field, the breaker wave height, $H_b$, and the corresponding wave angle, $\theta_b$, are extracted. The mean sea level, $z_s(x, y, t)$ is assumed to be uniform through all the domain except in the surf zone where a proxy for wave set-up is introduced (Ribas et al., 2023).

Given that Castell beach is an embayed beach, it is important to represent the wave shadow zones next to the lateral bound-

aries for off-normal wave incidence. This was not included in the previous versions of the model and has been specifically designed for this application. Following the overall rationale of the model (reduced-complexity), wave shadowing and diffraction by the lateral solid boundaries is treated in a simplified way. First, the wave field is computed as if the domain was open without solid boundaries. The "limiting wave ray", i.e., the wave ray just grazing the offshore tip of the up-waves solid wall, is determined. This defines the "shadow zone" as the area between this ray and the wall. The wave angles outside the shadow

zone are kept unaltered while the angles inside the shadow are approximated by an alongshore linear interpolation between the angle corresponding to the limiting ray and $0$ (shore-normal incidence) at the wall. The wave height computed by ignoring the walls, $H(x, y)$, is substituted in all the domain by $r(x, y)H(x, y)$, where $0 < r(x, y) \le 1$ is a factor representing wave diffraction. The Sommerfeld's solution for diffraction by a semi-infinite wall on a horizontal flat bottom (Dean, 1991; López, 2023) provides a proxy for this factor. It is $0.5$ at the limiting ray, it decreases towards the wall and rapidly increases to $1$ outside

the shadow zone. Outside the shadow zone, the values of $r(x, y)$ that according to the Sommerfeld's solution should slightly oscillate around $1$ are simply set to $1$.

### 3.4   Q2Dmorfo setup

The same computational domain of XBeach was used for Q2Dmorfo (Fig. 1) but with a different grid, $\Delta x = 5$ m, $\Delta y = 1$ m. The choice of the grid spacing was motivated by the horizontal length scale of the observed morphological changes in view

of previous applications of the model (see, e.g., van den Berg et al. (2012), Arriaga et al. (2017), Falqués et al. (2021)). The east and west lateral rocky headlands are represented by two rectilinear solid walls of $344$ m and $264$ m length starting at the $x$-axis, respectively. The time step was $\Delta t = 1.73$ s which is the largest value that ensures numerical stability. Regarding the boundary conditions, a zero sediment flux was assumed at the landward boundary and at the lateral boundaries representing the headlands which limit the embayed beach. The offshore boundary conditions were open, represented by a linear extrapolation

of the sediment flux. Finally, the wave and sea level data at the AWAC location (Sect. 2) were directly applied as boundary conditions at the offshore boundary of the domain, as in the XBeach case.





**Table 3.** List with several of the parameters values used in the Q2Dmorfo model.

| Parameter | Symbol | Def. value | Units | Range tested |
|---|---|---|---|---|
| Alongshore transport parameter | $\mu$ | 0.019 | $m^{1/2}\ s^{-1}$ | [0.016 - 0.022] |
| CERC additional parameter | $r$ | 2 | - | [0 - 3] |
| Cross-shore transport parameter | $\nu$ | 0.025 | - | [0.010 - 0.030] |
| Swash zone slope | $\beta_s$ | 0.16 | - | - |
| Equilibrium beach slope parameter | $D_1$ | 11.7 | m | [11.2 - 12.7] |

An important difference of Q2Dmorfo with respect to XBeach is that, for the former, an alongshore-uniform equilibrium beach profile must be defined (Eq. 8). A shifted Dean profile (Falqués and Calvete, 2005)

$$D(y') = B((y' + y_0)^{2/3} - y_0^{2/3})$$ (11)

is used, where $y'$ is the distance to the shoreline. The equilibrium bed slope $\beta_e = dD/dy'$ as a function of the water depth, $D$, is then extracted from this equation. The $B$ and $y_0$ parameters are computed from the slope at the coastline, $\beta_s$, and the depth $D_1$ at a distance $y' = 291$ m, which controls the overall slope of the equilibrium profile. In agreement with the observed bathymetry of January 2020, the shoreline slope was fixed to $\beta_s = 0.16$.

The most important parameters to be varied were those controlling the intensity of the alongshore transport, $\mu$, the intensity
of the cross-shore transport, $\nu$, and the equilibrium beach slope parameter, $D_1$. The tested rank and chosen default values are shown in Table 3. For the $r$ parameter in Eq. (7) the existing literature (Horikawa, 1988) advises $r \sim 1$ and we here examined values ranging $[0,3]$. Preliminary simulations proved that the best choice was $r = 2$.

## 4 Model calibrations

### 4.1 Metrics for the analysis

Both models were calibrated using the 6-month data set including two bathymetries and the wave and tide conditions measured in situ with the AWAC in the embayed Castell beach (Sect. 2). The models were initialized with the January 2020 bathymetry and the objective was to find the set of parameter values that provided the best model results compared with the observed bathymetry in July 2020.

To assess the performance of time evolution morphodynamic models, the Brier Skill Score (BSS) was used (e.g., Sutherland,
2004; Vousdoukas et al., 2011) since it measures the error in the model prediction relative to the observed changes.

$$BSS = 1 - \frac{\sum_N (Ymod - Yobs_f)^2}{\sum_N (Yobs_i - Yobs_f)^2}.$$ (12)

Here, N corresponds to the number of cells inside the area used to calculate the BSS, $Ymod$ to the final provided results by the model, $Yobs_f$ to the observed values in July 2020 (ground truth) and $Yobs_i$ to the initial values in January 2020. A BSS of 1



**Figure 4.** XBeach results obtained for the Brier Skill Score (BSS) metric of the coastline (a) and the bathymetry (b) and for the Standard deviation ($\sigma$) metric of the coastline (c) and the bathymetry (d) using all combinations of $f_{Sk}$ and $f_{As}$ parameters tested. The selected optimal parameter set is shown with a green dot in all panels. The default values shown in Table 2 were used for the rest of parameters.

 

means that the model perfectly reproduces the observed change, whereas a skill value smaller than 0 means that the errors in the
model prediction are larger than the observed changes. In van Rijn (2003) a classification was presented to assess qualitatively
the BSS values related to morphological changes (e.g., $0.3 < \text{BSS} < 0.6$ were considered "Reasonable" and $0.6 < \text{BSS} < 0.8$ were
called "Good").

In the Q2Dmorfo case, only the BSS of the coastline was computed because, regarding the bathymetry, Q2Dmorfo is in-
tended to resolve just the overall trends but not the details. Since XBeach was developed to simulate surf-zone morphody-
namics, the bathymetric BSS was calculated from the $-3.5$ m to the $0.5$ m, to embrace the areas with most significant bottom
changes. In addition, the XBeach coastline BSS was also computed in order to compare it with the Q2Dmorfo one. For each set
of parameter values tested during the XBeach calibration procedure, we performed 15 realizations to handle the randomness in
the offshore wave groupiness. Then, we computed a mean bathymetry out of these 15 realizations to finally calculate the BSS
of this bathymetry and its coastline. Also, one of our goals in the calibration procedure was to obtain an accurate but also robust
(i.e., reproducible) result, the standard deviation $\sigma$ between the results of the 15 realizations and the corresponding mean (of
both the coastline and the bathymetry) was also calculated to evaluate the potential dispersion within realizations.

For both models, the optimal set of parameter values were those providing a high value of the BSS, but for the XBeach
model a low value of the $\sigma$ was also required to ensure the robustness and repeatability of the results.

### 4.2 XBeach calibration

The XBeach model detailed calibration was performed by varying the key parameters on cross-shore sediment transport, $f_{Sk}$
and $f_{As}$ in the formulation of wave asymmetry (Eq. 2). All 49 combinations within the range of values for these parameters
shown in Table 2 were tested. The default values of the rest of parameters shown in that table were initially used. The values
$f_{Sk} = 0.55$ and $f_{As} = 0.35$ provided a high bathymetric and coastline BSS values and the lowest possible $\sigma$ values (Fig. 4).
Although the selected $f_{Sk}$ and $f_{As}$ values did not correspond to the highest possible BSS, these were high enough and, at
the same time, they provided the lowest variability within the 15 realizations. It was prioritized to have a robust and thereby
reproducible outcome. When smaller values of both parameters were tested, the BSS decreased because the modelled coastline
was seaward of the observed one and, in many cases, a negative value was obtained. This is due to the well-known overestima-
tion of erosion by the XBeach model when these parameters are close to their default value of $0.1$ (Kombiadou et al., 2021).
When larger values were tested, the BSS was also lower as the model, in these cases, underestimated the observed erosion and
the final modelled coastline was landward of the observed one. The $\sigma$ values increased when moving in any direction in the
parameter space (Fig. 4c-d).

Once the best pair of $f_{Sk}$ and $f_{As}$ values was determined, the bed friction coefficient $n$ and the acceleration factor $f_{mor}$ were
also varied. The value $n = 0.03$ m$^{-1/3}$ s was chosen for giving the highest BSS and the lowest $\sigma$ (Fig. 5). Lower values of $n$
induced higher erosion rates in the surf zone, while higher values prevented sand mobilisation in the nearshore zone, reducing
transport and erosion. Finally, the results were robust to changes in $f_{mor}$. No significant changes were obtained when values
of 5 or 20 were used, in agreement with Lindemer et al. (2010) and McCall et al. (2010).



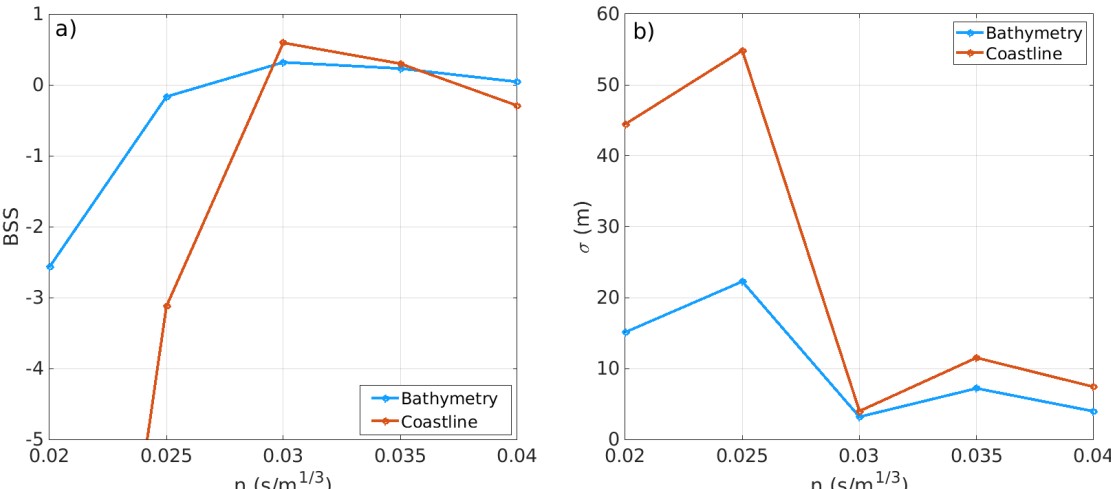

**Figure 5.** XBeach results obtained for the Brier Skill Score (BSS) metric (a) and for the Standard deviation ($\sigma$) metric (b) of the coastline and the bathymetry when varying the Manning coefficient $n$, using the optimum values $f_{Sk} = 0.55$ and $f_{As} = 0.35$. The default values shown in Table 2 were used for the rest of parameters.

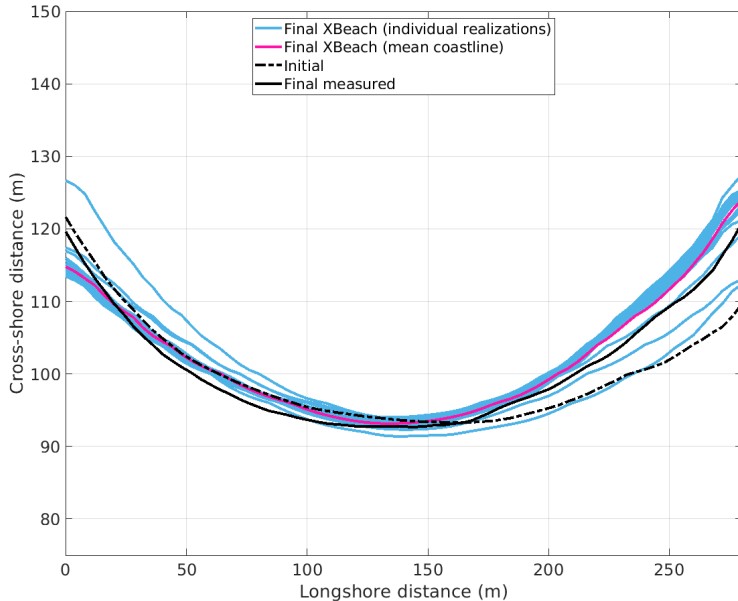

**Figure 6.** Measured coastlines in January 2020, (initial, dashed black) and in July 2020 (final, solid black), as well as XBeach modelled coastlines within the 30 realizations (light blue) and the corresponding mean (magenta). The default parameter values shown in Table 2 were used.





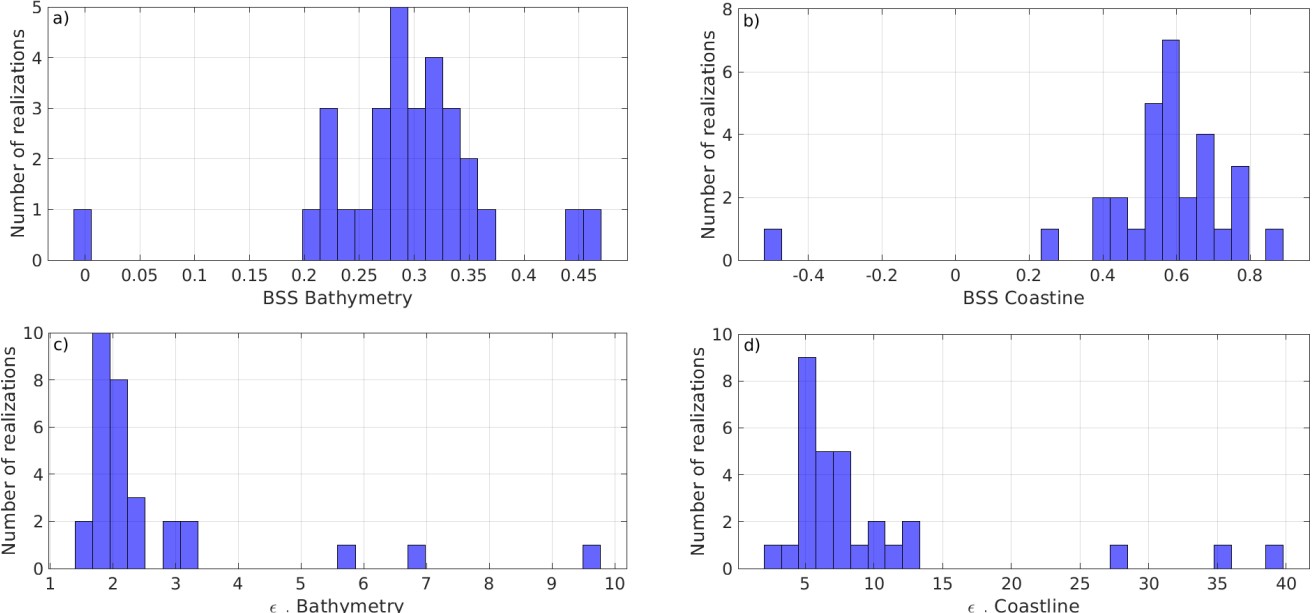

**Figure 7.** Distribution of BSS (panels a-b)) and $\varepsilon_{xi}$ (panels c-d) values of the bathymetry and coastline among the 30 realizations made with XBeach using the default parameter values (Table 2).

To ensure the robustness of the default case, 20 more realizations were performed only for the optimum parameter setting. Figure 6 displays the coastlines obtained within the 30 realizations (light blue), showing the low deviation between them and the computed mean coastline (dark blue). As can also be seen, the mean coastline and the majority of individual ones show a good performance in relation to the final observed coastline (dark solid line). The variability of the results of BSS and the root mean square deviation $\varepsilon_{xi}$ of the 30 individual realizations for the optimal set of parameters is also illustrated in Fig. 7. Numerous cases with high values of BSS and low values of $\varepsilon_{xi}$ were obtained, with a few of them giving low BSS values and a big $\varepsilon_{xi}$. These results show that the selected optimal values accomplish with the principles of robustness and repeatability that were targeted during the calibration procedure.

### 4.3 Q2Dmorfo calibration

The calibration parameters of Q2Dmorfo were the equilibrium beach slope parameter, $D_1$, and those controlling the longshore and cross-shore sediment transport $\mu$ and $\nu$, respectively. We tested 196 combinations within the range of values shown in Table 3. The best model performance (highest BSS) was obtained for $\mu = 0.019 \text{ m}^{1/2} \text{ s}^{-1}$ , $\nu = 0.025$ and $D_1 = 11.7$ m. As can be seen in Fig. 8, the BSS was very sensitive to $D_1$ which controls the overall progradation/retreat of the shoreline, low (high) values of $D_1$ producing shoreline retreat (progradation). For example, given a cross-shore bathymetric beach profile and for $D_1$ small enough, the equilibrium profile is shallower than the actual profile. In such situation, the actual profile (steeper than the equilibrium one) experiences an offshore gravitational transport that is more intense than the onshore wave driven





transport. Since the resulting sediment transport is seaward, the shoreline retreats and the actual profile tends to the shallower equilibrium one. The contrary occurs for large enough $D_1$. The $\mu$ parameter had less influence as can be seen from the overall

vertical trend of the isolines in Fig. 8. This was probably due to the long period (6 months) studied. During a particular storm, the shoreline of the embayed beach would tend to become perpendicular to the wave incidence direction. Whether this orientation is reached or not depends on a balance between the intensity of the sediment transport ($\mu$) and the duration of the storm. If the duration is long enough, the final shoreline orientation will be roughly independent of $\mu$. Here, given the long time period of the simulation, it turns out that the shoreline tended to a planview shape which was mainly determined by the

resulting mean wave direction, and the intensity of the longshore transport just influenced how fast this equilibrium planview was reached. It similarly occurred with the cross-shore transport, the parameter $\nu$ (which controls the time scale of the tendency to equilibrium) having even less influence than $\mu$. In the present long simulations, the final cross-shore bathymetric shape was mainly controlled by the prescribed equilibrium profile, being quite insensitive to the intensity of the transport ($\nu$).

## 5 Results

### 5.1 Morphodynamic evolution using in situ data

The calibration of the two models allowed simulating quite accurately the observed beach morphology after the 6-month study period. The BSS obtained for the XBeach optimum result was $0.38$ for the bathymetry and $0.74$ for the coastline (computed from the averaged bathymetry of 30 realizations). In the case of the Q2Dmorfo, the optimum simulation gave a coastline BSS of $0.79$ (Table 4). The bathymetric BSS (not used in the Q2Dmorfo calibration) was negative ($= -0.44$). According to van Rijn

(2003), the accuracy of the XBeach bathymetry simulation could be considered as "Reasonable" and the coastline simulation in both models would be "Good" (close to "Excellent").

The XBeach mean bathymetry (computed out of the 30 realizations) showed a good resemblance to the final bathymetry observed in July 2020 (Fig. 9). The XBeach model was able to simulate quite accurately the observed surf zone retreat from the shoreline up to 2 m depth but it predicted hardly no changes at larger depths. The Q2Dmorfo model was also good at

modelling the coastline but it was less precise in describing the surf zone bathymetry (Fig. 10, isobaths of $-1$ and $-2$ m). This is coherent with the fact that it is not designed to simulate the details of the bathymetric evolution (Sect. 3.3). However, the Q2Dmorfo bathymetric contours tended to qualitatively follow the observed changes in the $-3$ and $-4$ isobaths, except at the eastern side. In fact, a localised strong erosion (compared to observations) was produced by both models next to the eastern headland at depths larger than 2 m (Figs. 9b and 10b). Moreover, the models did not properly resolve the evolution of the dry

part of the beach, as the processes driving it were not included (role of the creek and eolian transport).

Both models simulated accurately the observed anticlockwise shoreline rotation (Fig. 11), consistent with an overall western directed sediment transport produced by the SE and SSE dominant wave incidence directions. XBeach tended to overestimate shoreline accretion during the 6-month study period, except at the easternmost zone. The shoreline simulated by Q2Dmorfo showed a too large retreat in the central part but in the western stretch of beach, which is the most exposed to the eastern

dominant waves and where more shoreline variability is observed, the adjustment between model and observation was very







**Figure 8.** Q2Dmorfo results obtained for the coastline BSS for all the combinations of $\nu$, $\mu$ and $D_1$ parameter values tested. The selected optimal parameter set is shown with a green dot in panel c.





**Figure 9.** Panel a: Comparison between the final bathymetry modelled by XBeach (red solid contours), the final observed one in July 2020 (black solid contours) and the initial one of January 2020 (black dashed contours and background colours). Panel b: Difference between the final modelled and observed bathymetries (background colours), with the modelled and observed bathymetric contours in red and black, respectively. The default parameter values were used (Table 2), with the wave and sea level measured by the AWAC.







**Figure 10.** Panel a: Comparison between the final bathymetry modelled by Q2Dmorfo (red solid contours), the final observed one in July 2020 (black solid contours) and the initial one of January 2020 (black dashed contours and background colours). Panel b: Difference between the final modelled and observed bathymetries (background colours), with the modelled and observed bathymetric contours in red and black, respectively. The default parameter values were used (Table 3), with the wave and sea level measured by the AWAC.





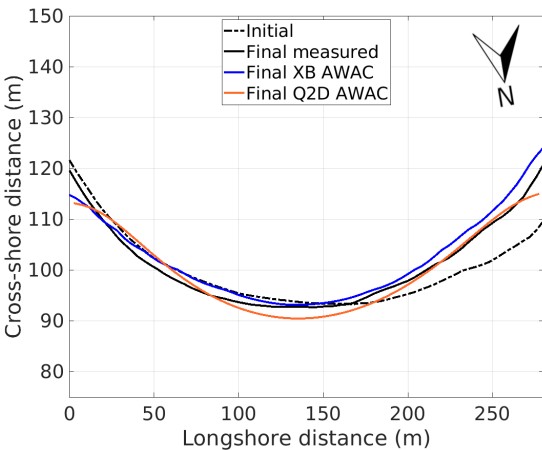

**Figure 11.** Comparison between the final modelled coastline using XBeach (solid blue) and Q2Dmorfo (solid orange) for the default parameter settings (Tables 2 and 3) and the wave and sea level measured by the AWAC. The initial and final measured coastlines are also displayed.

good. The westernmost and easternmost parts of the Q2Dmorfo modelled coastline experienced too much erosion, again due to the idealizations in modelling wave propagation with the rocky headlands.

## 5.2 Morphodynamic evolution using other forcing sources

To test the sensitivity of the modelled beach response to using other forcing sources, different combinations of the wave and
sea-level sources (described in Sect. 2) were applied using the parameters determined by the models calibration. Firstly, the AWAC wave data was combined with the 5-day averaged sea-level series measured by the same instrument, as well as with the Barcelona harbour gauge instantaneous and averaged series. Secondly, the wave data from the Cap Begur buoy propagated by SWAN was combined with the instantaneous and the 5 d averaged sea-level series from the Barcelona (BCN) harbour tide gauge. Finally, the wave data computed by CoExMed was combined with the instantaneous and averaged sea-level data from
the Barcelona harbour gauge, as well as with the instantaneous and averaged sea level from the CoExMed hindcast (see Table 4 for a list of combinations of the forcing sources). The default parameter setting resulting from the calibration (Tables 2 and 3) was used in both models. In order to add more robustness to the final results, a total of 30 realizations were carried out in XBeach for each combination of forcing sources tested.

Table 4 presents the BSS results obtained applying all the combinations of forcing sources in the two models. The simulations
with both models using wave data propagated with SWAN from the Cap Begur buoy gave a beach response similar than when using AWAC data but with a slight skill decrease. Essentially, the observed anti-clockwise rotation of the coastline was captured (Fig. 12). This is logical since the mean wave characteristics were similar to those of the AWAC wave series (Table 1). However, using the third source of wave forcing, the one from the CoExMed hindcast, significantly worsened the skill, obtaining negative BSS values in both models. The reason is that the CoExMed waves had angles biased towards the SW (Fig. 3 and Table 1).



**Table 4.** Brier Skill Score (BSS) from XBeach (XB) and Q2Dmorfo (Q2D) using the different forcing sources, where $\langle\rangle$ means a 5 d running average. The default parameter settings (Tables 2 and 3) were used.

| Wave source | Sea Level source | XB BSS bathymetry | XB BSS coastline | Q2D BSS coastline |
|---|---|---|---|---|
| AWAC | AWAC | 0.38 | 0.74 | 0.79 |
| AWAC | $\langle AWAC \rangle$ | 0.28 | 0.40 | 0.77 |
| AWAC | BCN | 0.42 | 0.67 | 0.79 |
| AWAC | $\langle BCN \rangle$ | 0.41 | 0.70 | 0.77 |
| BUOY + SWAN | BCN | 0.21 | 0.70 | 0.56 |
| BUOY + SWAN | $\langle BCN \rangle$ | 0.24 | 0.72 | 0.61 |
| CoExMed | BCN | -1.0 | -4.18 | -0.44 |
| CoExMed | $\langle BCN \rangle$ | -0.89 | -3.13 | -0.40 |
| CoExMed | CoExMed | -1.26 | -5.58 | -0.37 |
| CoExMed | $\langle CoExMed \rangle$ | -0.95 | -4.84 | -0.38 |

Then, both models underestimated the anti-clockwise rotation of the beach (Fig. 12) since there was less western directed sediment transport using this wave source.

There were no significant variations between the results obtained by the models using different sea-level sources when the wave source was maintained (Table 4). Also, the 5-day averaged sea-level series in general gave a result similar to the corresponding instantaneous sea level one. Exceptionally, using the AWAC averaged sea-level worsened the XBeach BSS values obtained using the AWAC instantaneous series (decreasing $\sim 30$ % and $\sim 50$ % the bathymetric and coastline BSS respectively), but the simulation skills remained Reasonable. No explanation has been found for the BSS worsening that occurs in this case.

To examine the modelled evolution of beach morphology in more detail, we defined a modified BSS (called $BSS^*(t)$ from now on) to account for time dependence. To do so, we applied Eq. (12) but with $Yobs_f$ being the result of the numerical run forced with in-situ AWAC measurements at every time step. In other words, the latter simulation is defined as the ground truth (or as the benchmark simulation) since it is the closest to the real changes (and used to calibrate the models). The advantage of this new metric, is that it allows evaluating the impact of the use of different forcing sources compared to the use of in-situ observations (Fig. 13f-h).

In both models, a similar morphodynamic response was observed with all the forcing sources during the first month, up to the storm in early March (the most energetic event of the entire study period, coming from the south). This strongest storm had a smaller effect when the AWAC source was used than when it was simulated using the other wave forcing sources. A pronounced decrease of BSS and BSS$^*$ was observed in both models, especially in those simulations using the CoExMed wave data. After this storm, there was a 15-day period of calm conditions with no major changes until another energetic period



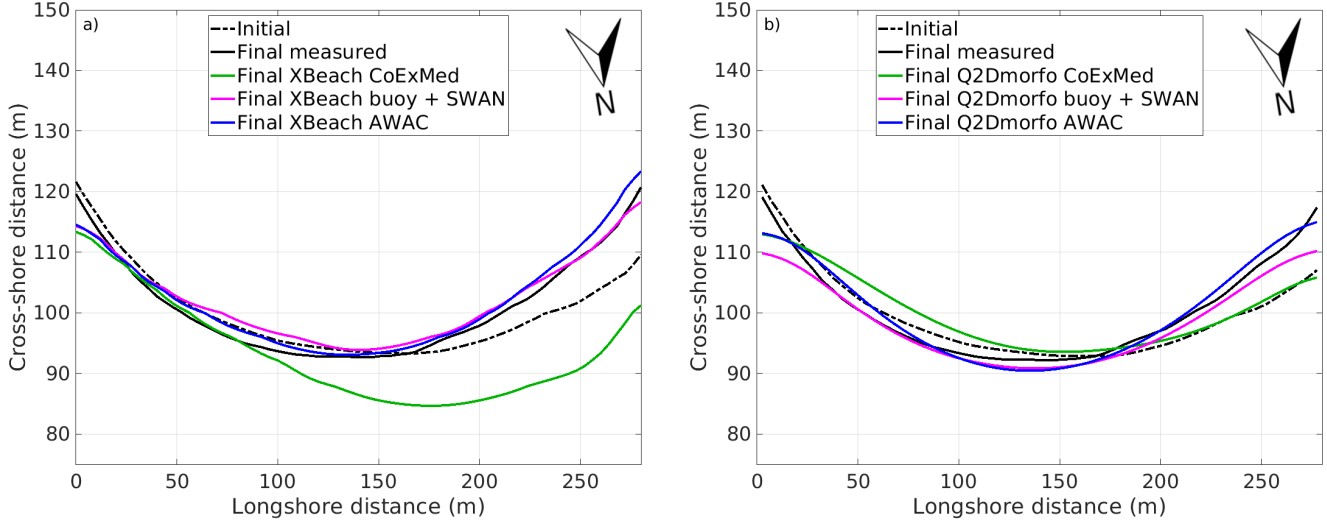

**Figure 12.** Final modelled coastlines using the three different wave forcing sources in XBeach model (panel a) and Q2Dmorfo model (panel b). In this figure, the sea level measured by the AWAC was selected for the AWAC wave source, the sea level from the Barcelona harbour was used with the buoy plus SWAN wave data and the CoExMed sea level was chosen for the CoExMed wave data. The default model parameter settings (Tables 2 and 3) were used.

of 1 month occurred, characterized by waves coming from the southeast. In the XBeach model, the BSS and BSS* values
increased in all simulations except for those using the CoExMed data. The Q2Dmorfo simulations during that episode tended
to have a similar behaviour for all combination of forcing sources obtaining analogous values of BSS and BSS*. During the
last 2 months, a combination of calm and moderate conditions reached the beach with waves alternating between south and
southeastern directions. These conditions affected the beach similarly in both models, with a generalized decrease in BSS and
BSS* when the CoExMed data was used. The behaviour obtained when the data propagated from the buoy was used was
similar to that of the in situ data.

## 6   Discussion

### 6.1   Optimum model setup and parameter values

Simulating the morphodynamic beach response of Castell beach over a 6-month study period using XBeach was a significant
challenge because this model is typically applied to shorter time scales, from days to weeks. Using the `surfbeat` mode,
enabling the random mode (`random = 1`) and performing many realizations (15-30) of each simulation (as described in
Sect. 3.2) allowed us to reproduce the uncertainty and variability of real stochastic wave climates within XBeach simulations.
This resulted in more reliable and realistic outcomes, giving significantly high values of the BSS in one of the few successful



**Figure 13.** Time evolution of the $BSS(t)$ during the 6 month study period, calculated with Eq. (12) using the time varying XBeach modelled bathymetries (panel c), XBeach coastlines (panel d) and Q2Dmorfo coastlines (panel e) and the corresponding final measurements as ground truth, for all the combinations of wave and sea-level forcing sources. Also, the time evolution of the $BSS^*(t)$ during the 6 month study period, calculated with the instantaneous bathymetry and coastline from the simulation forced with AWAC data as ground truth and using the time varying XBeach modelled bathymetries (panel f), XBeach coastlines (panel g) and Q2Dmorfo coastlines (panel e), for all the combinations of wave and sea-level forcing sources. The time evolution of $H_s$ (panel a) and $\theta_m$ (panel b) for the three wave forcing sources are also shown.





applications of XBeach to a 6-month period. The implemented methodology is in line with that of Rutten et al. (2021), who also demonstrated the importance of including the wave time series randomness in XBeach simulations to accurately model bed
evolution response, particularly in the complex and dynamic nearshore zone. This is an important learning for future XBeach studies that intend to simulate time periods longer than a week or so. The approach followed in the present study was highly time consuming and involved extracting the mean bathymetry and its shoreline from the 15-30 realizations for each parameter setting and for each hydrodynamic forcing source combination. Thereby, it required a long and iterative calibration procedure to finally find the optimal parameter values.

In agreement with our results, previous studies also showed that increasing the wave skewness and asymmetry (`facSk` and `facAs` factors) lead to an increase of the onshore sediment transport and mitigate the well-known issue of erosion overestimation in XBeach simulations. For instance, Schambach et al. (2018) demonstrated that rising these factor values above their default setting ($0.1$) resulted in an improved performance, with an optimal value of $0.3$ for both parameters in the analysis of cross-shore profile evolution during a storm in an open beach in Rhode Island. Similarly, Kombiadou et al. (2021) used higher
values ($0.65-0.75$) to reduce the erosion overestimation in cross-shore sections during storm periods in a 2-month simulation on Faro Beach, an Atlantic open beach in South Portugal. Furthermore, Sanuy and Jiménez (2019) conducted an extended calibration of these parameters to simulate a stormy period in an open beach in the Catalan coast, identifying an optimal value of $0.6$ for each factor. Remarkably, the optimum values obtained in this study (`facSk`$=0.55$ and `facAs`$=0.35$) are consistent with those reported previously. In fact, as shown in Fig. 4, positive values of the BSS (dark red) were only obtained for high
values of these two parameters. Notice that, since the first topo-bathymetry in January 2020 was measured a few days after the Gloria storm, which was the strongest in at least 30 yr and probably induced a significant beach erosion, such large values of `facAs` and `facSk` were probably needed to compensate the potential storm-induced erosion with an increasing onshore transport. Using the `wave_averaged` mode on the `turb` parameter showed good results mitigating the beach erosion observed when the default mode (`bore_averaged`) was used. Previous studies such as Kombiadou et al. (2021) also used this
mode obtaining good outcomes with a realistic erosion trend compared to the observed data. The simulations to assess the optimum value of the Manning bed friction coefficient ($n = 0.03$ m$^{-1/3}$ s, Fig. 5) revealed its influence on the model performance. Similar findings were presented in Melito et al. (2022), where the importance of this parameter was also highlighted, emphasizing the requirement of increasing its default value (from $0.01$ m$^{-1/3}$ s to $0.045$ m$^{-1/3}$ s).

The Q2Dmorfo skill to model coastline behaviour is also noteworthy, bearing in mind the amount of idealizations behind
this model. This positive result proves that the present model version is appropriate for embayed beaches. In fact, when the default simulation was repeated switching off the recently included effect of the headland's shadow on the waves (described in Sect. 3.3), the model results became completely unrealistic compared with the observations. The most critical Q2Dmorfo parameter was $D_1$, controlling the overall slope of the equilibrium profile. The obtained best value ($D_1 = 11.7$ m at 293 m from the shoreline) gave an equilibrium profile that was consistent with the overall trend of the first 6 m depth of both bathymetries used in the calibration. In other words, the equilibrium profile selected by the calibration follows the observed bathymetries
within the upper shoreface, the most active area. Interestingly, the selected equilibrium profile fits somewhat better the final bathymetry (see the dashed line in Fig. 2b). This is likely due to the fact that the initial one was taken just after the Gloria storm





so that the beach was probably a bit far from equilibrium at that time. The optimum values of the sediment transport parameters in Castell beach ($\mu = 0.019$ m$^{1/2}$ s$^{-1}$ and $\nu = 0.025$) were half the ones obtained in the detailed Q2Dmorfo validation with

data from the Sand Engine, the Netherlands (Arriaga et al., 2017; Ribas et al., 2023). This is not surprising because the grain size of the study site is $50\%$ larger than the one at the Dutch coast and the water velocities are smaller due to the embayment influence, both factors resulting in lower sediment transport rates. Notice that the value of the $K$ parameter in the CERC constant corresponding to $\mu = 0.019$ m$^{1/2}$ s$^{-1}$ is $K = 0.065$, smaller than the lowest values found in the literature. However, there is a high uncertainty regarding the $K$ value (Arriaga et al., 2017) and the present detailed study is a good opportunity

to assess it in embayed beaches, which had been scarcely modelled before. To confirm the article findings, the calibration procedure of Q2Dmorfo was also pursued using CoExMed forcing for both waves and sea level. The obtained optimum parameter values were the same as for the AWAC forcing calibration but the skill was negative, $BSS = -0.37$. Interestingly, by playing within a wide range of $D_1, \mu, \nu$ parameters there was no way to improve this skill. This is important since it shows that the good skill obtained when forcing with AWAC is not an artifact of the parameter selection, but has to do with the physics

included in the model.

## 6.2    Comparison between the performance of the two models

Despite both models provided a good prediction of the beach evolution during the 6-month study period, discrepancies were observed when comparing their results to the final observed topo-bathymetry. Both models presented a remarkable eroded area at the easternmost part of the beach at depths of approximately 3-4 meters (Figs. 9b and 10b). A probable explanation for this

issue could be the oversimplifications employed by both models to represent the real behaviour of waves as they propagate towards the coast from the southeast and interact with the headland. This is much more noticeable in the Q2Dmorfo case, which shows larger model-data differences and these extend to deeper waters (Fig. 10b), and happens because this model is significantly more idealized (see Sect. 3.3). In particular, the simplifications affecting the easternmost side are: i) assuming monochromatic waves that then form a sharp shadow zone, ii) neglecting the role of the surf-zone currents (and bars) that

might play a role near the headland and, most importantly, iii) using a simplified cross-shore sediment transport based on an imposed alongshore-uniform profile whilst measured bathymetries are shallower in this easternmost area compared with the rest of the beach (as can be seen in the first $40$ alongshore meters in Fig. 10a). These idealizations are an important factor to explain why bathymetric BSS in Q2Dmorfo always had negative values. In fact, when the bathymetric BSS is calculated in both models deleting the first 40 meters in the eastern part of the beach, the values obtained significantly increase ($\sim 200\%$ in

Q2Dmorfo and $\sim 40\%$ in XBeach). In Q2Dmorfo, the BSS obtained reached $0.43$, whereas XBeach obtained a BSS of $0.52$. Additionally, the complexity of the real shape of the rocky headland, which is represented by a simple rectilinear wall in the Q2Dmorfo model and by a 2x2 non-erodible pillar in XBeach, also contributes to the differences at the easternmost side in both models. Finally, since neither model simulates the dry beach, there were big differences in that region between model results and the topo-bathymetry of July 2020. Processes not included in the models, such as the movement of the stream mouth, its

discharge during rainy periods and the eolian action, contribute to these differences.



To assess how the models differed on their morphodynamic response throughout the 6 months, the bed level and shoreline variabilities were calculated in the two simulations forced by the in situ measurements from the AWAC (Fig. 14). The alongshore-averaged shoreline variability was defined as

$$\langle \Delta y_s(t) \rangle = \left( \frac{1}{L_x} \int_0^{L_x} (y_s(x, t+\Delta t) - y_s(x,t))^2 \, dx \right)^{1/2} \tag{13}$$

with $\Delta t = 12h$. A similar expression was used for the surface-averaged bed level variability, $\langle \Delta z_b(t) \rangle$ (involving $z_b(x,y,t)$ and the integral being in $x$ and $y$). An important contrast was observed between the models in the bed level variability during the first month, where Q2Dmorfo showed significantly greater changes than XBeach. This strong Q2Dmorfo variability was induced by the model tendency to reach the same imposed equilibrium profile all along the beach and, in particular, at the easternmost section. The equilibrium profile shape arising from the calibration was consistent with the overall trend of

both measured bathymetries (see Sect. 6.1) and was assumed to be alongshore uniform. However, the measured bathymetries clearly showed shallower-than-average profiles in the easternmost 40 m along the beach. Thereby, the initial storms produced fast and substantial changes in the modelled easternmost area to reach the equilibrium shape. Throughout the next 2 months, which included the strongest storm and subsequent eastern-dominated wave conditions, both models showed similar bed level variability, with significant changes during the high energy events and minimal changes during calm periods. Along the last

2.5 months, the bed level changes in Q2Dmorfo were again larger than those of XBeach, particularly during storms. Regarding the shoreline variability, both models presented a similar behaviour during the 6 month period (Fig. 14d) but XBeach generally produced higher changes than Q2Dmorfo, i.e., the shoreline reacted quicker to storms in XBeach than in Q2Dmorfo. The probable reason is that the differences between the idealized cross-shore transport in Q2Dmorfo and the more realistic description by XBeach become more pronounced in very shallow water. Finally, it is interesting to note that despite Q2Dmorfo coastline

responds less to individual storms than XBeach coastline it eventually reaches the same values in the medium term.

### 6.3 Implications of the assessed role of the forcing sources

The results obtained using the different wave and sea-level forcing sources emphasize the importance of having a good description of the wave mean direction (Sect. 5.2), particularly for simulating the morphodynamic response of an embayed beach such as Castell beach. The simulations using CoExMed wave data, which contain a bias in wave angle (Table 1), could not reproduce

the observed rotation of the shoreline during the study period (Fig. 12). This effect was magnified when the XBeach model was used, as it resolves more processes compared to the more simplified approach of Q2Dmorfo, and is then more sensitive to the wave conditions. The $BSS(t)$ using the various forcing sources did not differ much during the first month (Fig. 13). However, the early March storm had varying effects on the beach morphology depending on the wave forcing source used. When the waves from the AWAC were used, the coastline BSS increased during the storm, especially for Q2Dmorfo, meaning that the

beach evolved towards its final configuration, while the XBeach bathymetry BSS slightly decreased. The wave conditions obtained by propagating the buoy data with SWAN produced a modest shoreline BSS increase with Q2Dmorfo and a decrease for XBeach. However, the BSS converged with that corresponding to the AWAC data forcing during the following storm (showing



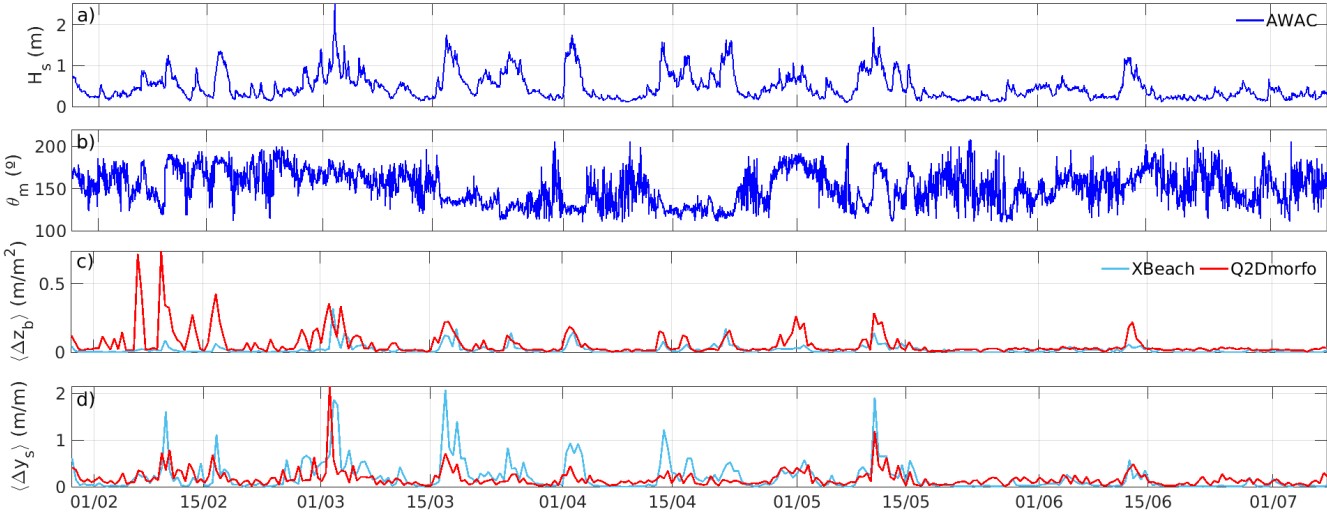

**Figure 14.** Differences in the instantaneous modelled bed level variability during the 6 month study period, when both models were forced with the AWAC data. The time evolution of $H_s$ (panel a), $\theta_m$ (panel b), bed level variability ($\langle \Delta z_b(t) \rangle$) (panel c) and shoreline variability ($\langle \Delta y_s(t) \rangle$) (panel d) for the two models are shown.

high BSS and BSS* values at the end of March, see Fig. 13). At the end of the study period, a beach response comparable to that of AWAC simulation was also obtained (Fig. 12), providing only slightly smaller values of BSS and BSS*. The results obtained with the CoExMed wave data showed the worse behaviour, particularly after the early March storm, which eroded the beach more than using the other forcing sources. The BSS and BSS* never converged back to the values of the AWAC simulation and at the end of the study period they were always negative. This indicates that, when forced with CoExMed wave data, the beach was not able to recover from the erosion suffered during the energetic episode and could not rotate properly during the rest of the time period.

To understand to what extend this early March storm was the turning point that led to significant differences between the results obtained using the CoExMed wave conditions, two additional simulations were conducted. Firstly, the modelled bathymetry from the simulation forced with AWAC data (both waves and sea level) in both models was extracted on 15 March, i.e., about a week after the storm to allow the XBeach bathymetry to stabilise (this model typically produces numerical noise during storms). These bathymetries were then used as initial conditions to simulate the remaining 4-month period using the CoExMed data forcing. The same procedure was also applied but reversing AWAC and CoExMed input data. To compare the results and get further insights into the role of the forcing sources, the BSS* metric defined in Sect. 5.2 was again evaluated for these two additional simulations (Fig. 15). The original run forced with CoExMed data during the 6 months is also shown for comparison. Notice that the BSS* metric uses the simulations with 6-months AWAC data as ground truth (hence assuming




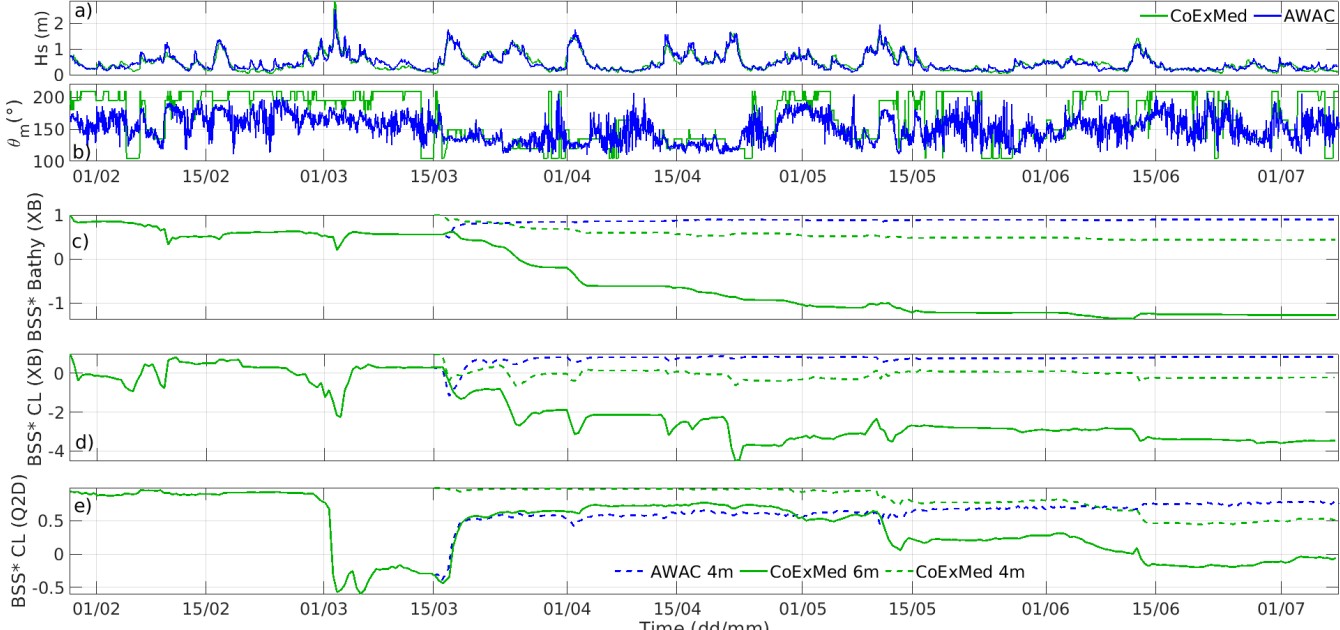

**Figure 15.** Time evolution of the cross-simulations BSS*, calculated with the instantaneous bathymetry and coastline from the simulation forced by AWAC data as ground truth and using the time varying XBeach modelled bathymetries (panel c), XBeach coastlines (panel d) and Q2Dmorfo coastlines (panel f). The time evolution of $H_s$ (panel a) and $\theta_m$ (panel b) for the two wave forcing sources are also shown.

it is the most realistic) so that BSS* quantifies how a simulation with another forcing source diverge from the one forced by in
situ data.

Despite starting with a more eroded bathymetry caused by the CoExMed data of the early March storm, when we subsequently applied the AWAC data the beach was able to recover and simulate the observed final shoreline rotation in the two models (see the dashed blue lines in Figs. 15c-e, with final BSS* close to 1). This can be compared to the 6-month simulations forced with AWAC data that correspond to $BSS^* = 1$ throughout the whole period. In contrast, when the more realistic
bathymetry obtained in 15 March with the AWAC forcing was subsequently simulated with the CoExMed data for the remaining 4 months the errors in the latter source kept producing accumulated differences in the modelled morphology and gave worse final values of BSS* (green dashed lines in Figs. 15c-e). However, results were better than those obtained when the CoExMed data was applied for the whole 6 months (green solid lines). This indicates that the obtained discrepancies when the CoExMed data source was used are partially attributable to the early March storm and also to the errors throughout the whole
data series. This highlights the importance of having accurate wave data series not only during the storms but also during the rest of the time. On the other hand, our results also indicates that if a wrong data source is used for a short period (i.e., in our case, 2 months) but a more accurate data source is applied afterwards, the morphodynamic model simulations can partially recover their reliability.





The most important implication of this study is that using different wave data sources critically modified the outcome of the morphological simulation. In particular, the known errors in wave direction of existing wave hindcasts of the Spanish Mediterranean coast (shown in Fig. 3 for the CoExMed hindcast and in De Swart et al. (2021) for other existing hindcasts) can produce completely unrealistic morphological simulations. This might be especially important in embayed beaches where the waves interact with the structures that limit them and the wave direction is modified due to all the intrinsic propagation processes. Our recommendation for long-term studies is to use the nearest wave buoy and carefully propagate to the site the measured conditions during the study period (see De Swart et al. (2021) and the Supplementary Information for more details on the proposed methodology). However, buoy data contain gaps that are often filled in with hindcast data. The above discussion about the results obtained in the present study when combining these two types of wave source conditions (Fig. 15) underlines that a wrong result produced by errors in a wave data source during time periods of the order of 1-2 months can be compensated if a correct data source is subsequently applied. An alternative to improve the hindcast data accuracy and thus, the results obtained, could be a previous calibration or a bias correction of the hindcast wave direction. Also, long-term hindcasts can be very useful to fill in the wave buoy gaps with more sophisticated data imputation techniques. In any case, since these results could be site dependant, it is advisable to perform tests of the sensitivity of morphodynamic modelling to the forcing conditions such as the one presented here before performing long-term studies.

The effect of the choice of sea level data source was much less important than that of the wave source (Table 4). For example, by comparing the instantaneous data series and the 5-day filtered data series in the 6-month study period, no significant changes were observed (with the only exception mentioned in section 5). This could be attributed to the fact that Castell beach has a very small tidal range and, thereby, the differences between the instantaneous and the filtered data series were not substantial enough to result in significant changes in the beach response. The implication of the minor influence of the chosen sea level data source is that different available long-term sea level data sets can be used when simulating the long-term beach morphological evolution, including tidal gauges located in harbours at distances from the beach of the order of 100 km (such as the Barcelona harbour gauge in the present site). In any case, the choice of sea level source could be more influential in beaches with larger tidal range.

# 7 Conclusions

The morphodynamic evolution of the embayed beach of Castell (northwestern Mediterranean Sea) during 6 months has been successfully reproduced using two different morphodynamic models, the 2DH XBeach and the reduced-complexity Q2Dmorfo. Remarkably, despite XBeach was designed to specially simulate storm episodes, very realistic outcomes compared with observations have been obtained in the present longer-term simulations after calibrating it with in situ data. The following ingredients are essential to avoid erosion overestimation in such type of medium term XBeach simulations: including the randomness of wave groupiness present in real beaches, performing tens of realizations to account for such randomness, and selecting appropriate values of the cross-shore sediment transport and bed friction parameters. It is important to note that the topobathymetry obtained in January 2020 (used as the initial bathymetry for the models) was obtained a few days after the Gloria storm. It



probably affected the beach morphology, which had to recover at the beginning of the study period. This could be one of the main reasons for the high values of the cross-shore transport parameters obtained in the XBeach calibration. Moreover, even though the Q2Dmorfo model is significantly simpler because it was designed to simulate the shoreline evolution over
decadal temporal scales and despite it does not respond accurately to individual events, it has provided excellent results during the 6-month period after calibration. So, this confirms that this model is appropriate to simulate the seasonal morphodynamic evolution of embayed beaches.

The choice of the wave forcing source can significantly affect the accuracy and reliability of the results of both types of models. The effect is stronger in XBeach because it includes more physical processes and simulates stronger changes, like
those produced by individual storms. In both models, the simulations using the propagated data from the buoy (using SWAN model) provide results quite consistent with those using in situ data (AWAC). In contrast, those obtained with the hindcast data (CoExMed) exhibit greater discrepancies mainly due to the existing bias in wave direction. These inaccuracies are present throughout the full hindcast data set and produce model errors that accumulate in time, the modelled coastline being unable to rotate as in the observations. Interestingly, even after recalibrating the Q2Dmorfo using the hindcast wave and sea-level data
series, poor values of BSS are obtained since it is not possible to reproduce well the observed shoreline rotation. This shows that the good skill obtained by using in situ data has to do with the physics in the model rather than being an artefact of the parameter selection. On the other hand, the accuracy of the present simulations hardly depends on the sea level data source, even if tides are filtered, probably because they are small on many Mediterranean beaches.

This study shows that accurate wave information is fundamental in morphodynamic modelling to capture the complex
dynamics of beach morphology, including shoreline changes and erosion processes. As an alternative to in situ data, propagated waves from nearby buoys can be used. Inaccurate wave data that are often present in existing hindcasts, especially regarding wave direction, may lead to unreliable predictions of beach evolution, particularly in embayed sites. Hindcast data, however, can still be a useful option to fill in gaps in buoy data, especially if correction algorithms are implemented for the direction bias. Overall, this study indicates the importance of using realistic forcing sources for long-term morphodynamic projections
in the context of climate change modelling.

**Data availability**

The codes and data supporting all results showed in the manuscript are available from the corresponding authors upon request.

**Author contributions**

NCB, AF, FR and DC planned and designed the idea of the study. NCB and AF carried out the models simulations, which were
previously designed and subsequently analysed along with FR and DC. The two topobathymetries were obtained and processed by RD, CMP and AFM. The AWAC data was obtained and provided by AFM, the CoExMed data was computed by MM, AA and TT, and the data from the buoy was propagated by RdS. The paper was written by NCB with important contributions



from AF, FR and DC, and it was revised by all the other co-authors. All authors approved the final version of this manuscript. Financial support was obtained by AF, FR and JG.

**Competing interests**

The authors declare that they have no conflicts of interest.

**Acknowledgements**

This study was supported by grants RTI2018-093941-B-C33, PID2021-124272OB-C22 and TED2021-130321B-I00, funded by MCIN/AEI/10.13039/501100011033/ of the Spanish government and by "ERDF A way of making Europe".





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
