# Peer review of "Role of the forcing sources in morphodynamic modelling of an embayed beach"

_EGUsphere, 2023_

## Author Response (AR1)

**We have received two positive reviews on your manuscript, both providing helpful comments and suggestions for further improvements. Please proceed with a minor revision of the paper, particularly bearing in mind the overall recommendation from reviewer #1 to consolidate the work into a somewhat more compact presentation, by transferring some of the more technical details into Supplementary Materials.**

We thank the two anonymous reviewers and the handling editor for the constructive and helpful comments on our manuscript. We implemented most of their suggestions, which improved the document. In this letter the detailed response to all the reviewers' comments can be found, where we also underline the sentences describing the specific changes implemented in the article and indicated the location of that changes in the version of the marked-up manuscript version of the paper.

**Reviewer #1**

**The paper is well written, interesting and quite informative as it allows the reader to understand the different steps of the work and follow even technical details. So overall I am positive, I have just some suggestions that the authors could consider as I believe they could improve the paper.**

We thank the reviewer for the constructive and helpful comments on our manuscript. We have  implemented most of their suggestions, which improved the document. Below we respond to all comments. We underlined the sentences describing the specific changes implemented in the article and indicated the location of that changes in the version of the marked-up manuscript version of the paper.

**The authors collect field data of waves, water levers and topography-bathymetry and after a thorough calibration effort, they apply 2 different models to simulate long term shoreline evolution. This involves a substantial amount of work, but in order to result in a meaningful journal paper, the authors need to focus on the information that is more of interest to the reader, otherwise the main message is lost inside the large amount of technical information. Now the manuscript feels a bit like a thesis or technical report and does not convey clearly the important findings. So my comments are mostly to improve this aspect, since the manuscript now is rather long and tedious for the reader (33 pages, 15 figures). I believe that a leaner version with the most important information could make it more friendly. Several technical aspects could be moved to appendices, with such examples being the description of XBeach and the model setup (so that XBeach and Q2Dmorpho sections could become of similar size).**

We agree with the reviewer and editor suggestion and we have moved the description of all the equations of XBeach and Q2Dmorfo to the Supplementary Information in order to reduce the paper length. In the main manuscript we have only left the information that is needed to understand the simulations of this article but without going deep into the model details, as the reviewer and editor suggest. To shorten even more the main text, we have also moved

figures 4, 5 and 8, relative to the models calibration sections, to the Supplementary Information.

*Changes on Sect. 3 and 4, pages 8 to 17, and in Supplementary Information, Sect. C and D.*

**I highlight below what are the key findings in my opinion…**

**The authors have put substantial effort to calibrate XBeach and Q2Dmorpho and this is an important aspect of the paper. I believe that the abstract and the paper highlights (if the journal offers this possibility) should highlight the % improvement in BSS compared to the default calibration. I find this the most important finding of the work (and potentially should feature also in the title) and one that could incentivize potential readers to read the entire paper which is rather long and provides a lot of information.**

We agree with the reviewer that we should give more importance to the calibration outputs, and at the end of Sect. 6.1 we have included that there is a 65-85% decrease in the error of the calibrated results compared with those obtained using the model default settings. We have also added this information in the abstract and conclusions. However, we consider that this is too detailed for the title of the study. The main objective of this study is the comparison between the three forcing sources to identify which one is the best one to do long-term simulations in the climate change context, as the reviewer explained later.

*Changes on the Abstract (page 1, lines. 6-7), Sect. 6.1 (page 28, lines 608 to 614) and in the Conclusions (page 33, lines 736-737).*

**The aspect of the forcing conditions is more straightforward. It is not surprising that SWAN modelling can serve as a good 'transfer function' of buoy measurements closer to the beach and that the reanalysis will result in more epistemic errors. In my opinion the key point here is to highlight that % of errors that such large scale reanalyses (or projections) introduce, because most likely they are going be mainly used to produce hindcasts of forecasts of shoreline change. So it is important that the community have quantitative information of the uncertainties introduced by the use of such reanalyses.**

We agree that we might quantify and emphasize more the errors induced by the large-scale reanalysis as it is what is going to be used in long-term simulations. In Sect. 6.3 we have included that there is a 314% and a 81% increase in error using the buoy propagation data against the hindcast data to simulate the coastline evolution in XBeach and Q2Dmorfo respectively.

*Changes on Sect. 6.3 (page 32, lines 709 to 711).*

**Another important finding is to what percentage the use of Q2Dmorpho affects the final BSS compared to XBeach and how computational times compare… What is the % reduction in computational effort and is there a real motivation to use simpler models? Are there limits to the latter's application? Is there a time frame after which the deviations become too high, rendering the simulations not meaningful?**

In Sect. 6.2, we have added a description of the differences in terms of computational effort when the simpler model is used. The Q2Dmorfo model is 500 times faster than XBeach so that it can be really useful to model long time scales. In different places, we have mentioned that the most important limitation of the Q2Dmorfo model (the "price" paid) is that it simulates well the coastline evolution but the bathymetry is not accurately represented. We have not compared how these two models perform in time scales longer than 6 months, but in a previous application of Q2Dmorfo model we proved that it can accurately reproduce the evolution of a sandy coast during 8 yr (Ribas et al., 2023).

*Changes on Sect. 6.2 (page 30, lines 657 to 662).*

**Topobathymetric data. Normally equilibrium profiles (like the ones shown in Fig 2) should be compared to longshore averaged profiles measured on site. The authors should justify why they used data only from one transect, why the lower submerged part deviates from the theoretical one. Do such deviations affect the output of the simulations?**

The equilibrium profile depends on two parameters, the slope at the shoreline $\beta s$ and the overall slope parameter, D1. The former is fixed to 0.16, but the latter is a calibration parameter ranging from 11.2 to 12.7 m so that the equilibrium profile is not fixed. We are now more explicit on this in Sect. 4.2 of the article. The aim of Fig. 2 is a rough comparison between representative observed profiles and a representative equilibrium profile. We agree with the reviewer that the best option to represent the observed profiles is their alongshore average. Therefore, in Fig. 2 we now display them instead of the central profiles, both in January and in July 2020. Regarding the equilibrium profile, we now show the full range of explored equilibrium profiles along with the optimum one from the calibration (D1=11.7 m). We do think that this new Fig. 2 is much more illustrative, in line with the reviewer's comment.

From -8 m until -16 m the equilibrium profiles are significantly shallower than the measured bathymetries, as the reviewer mentions. This is because the Castell beach bathymetry in deep water is quite planar rather than concave up as in the Dean profile. However, such deviation in the lower submerged part does not play any role because in these water depths there is no wave stirring and thereby the sediment transport is insignificant. Consistently, the model keeps the initial bathymetry in this region almost unchanged after the half-year wave forcing, despite being relatively far from the equilibrium profile. We have added a sentence with a comment about this in Sect. 6.1.

*Changes on Sect. 4.2 (pages 17 to 19), Sect. 2 (page 5) and Sect. 6.1 (page 28, lines 591 to 593).*

**Lines 400-415: From one hand I see in Figs 10-11 that the shoreline is slightly curved, but if I understand well the authors try to explain that the model tends to produce a straight shoreline and I find it rather confusing why this is the case. One issue I can imagine is that the use of the same profile across the same beach is not realistic since the dominant longshore transport direction would result also in alongshore profile differences (including slope gradients which seem to be important). Still, the refraction, especially along the hard boundaries of the beach, should result in alongshore differences of Hb which should result in alongshore morphological differences. Shouldn't the shadowing also apply to the longshore current?**

The observed shorelines are indeed slightly curved and so are the modeled shorelines. In line with the reviewer comment, this curvature is the result of alongshore differences in Hb and wave angle. These differences are partly forced by refraction and shadowing by the hard boundaries and partly self-organized by the feedback between the waves and the developing morphology. We have included a modification in the text in Sect. 4.2 to avoid confusion. There we have explained that the overall trend of the (not straight) shoreline, including that curvature, was rotating according to the dominant overall wave direction.

*Changes on Sect. 4.2 (page 19).*

**This brings me to another point related to the way refraction is dealt in the model. How can the refraction calculations be accurate without considering the solid boundaries? Doesn't the bathymetry change along the boundaries? Shouldn't waves refract along the headland? I think that this part could be better described in an appendix with a figure of the wave rays.**

The two models consider wave transformation by the rocky headlands in a different way.

Regarding Q2Dmorfo, for every wave angle at the tip of the (waveward) headland, a shadow zone next to the headland is defined by a limiting wave ray. Inside the shadow zone, wave refraction and diffraction is considered parametrically somehow imitating Sommerfeld's solution, so that there is indeed wave refraction. There are also gradients in sediment transport thereby bathymetric changes. All this is now described briefly in Sect. 3.2 and in more detail in the Supplementary Information Sect. C2.

Regarding the XBeach model, refraction is included inside the detailed wave transformation module and it is calculated over the full evolving bathymetry. Wave shadowing and diffraction by the lateral solid walls are also considered by imposing non-erodible pillars at the location of the offshore tip of the headlands, These pillars influence wave propagation from the offshore boundary to the coast, generating the proper wave shadowing and diffraction due to the presence of rocky headlands and avoiding the typical scour effects of placing rectilinear solid walls in this model. We have clarified this in Sect. 3.1.

To visualize these effects, we plot here an example of the wave height (in colours) and direction (with vectors of the wave number) as simulated by the XBeach model (left) and Q2Dmorfo model (right).

[Figure]

These figures do not correspond to the final calibrated result but to more idealized simulations but they perfectly illustrate how waves propagate near the headlands. This article is already very long and detailed, as the reviewer says, and we consider that it is not necessary to add these types of figures.

*Changes on Sect. 3.2 (page 14, lines 356 to 366), Supplementary Information on Sect. C2 (page 8) and Sect. 3.1 (pages 10 and 11, lines 231 to 276).*

**Anyway, I don't think the word 'planview' is the best option here and I also believe that this discussion should be placed later, when the results of the model are presented.**

We have deleted this word since it could indeed be confusing.

*Changes on Sect. 4.2 (page 19, lines 477 and 478).*

**There is an issue that has been already discussed but maybe deserves more discussion. The simulation period is after a major storm and this could potentially imply that the calibration could be more biased towards accretive conditions than if for example the start was 1 month before. Given that there are no interim surveys to**

**validate how the model performed during the simulation period and the validation is based only on the initial and final topography, that could be an issue.**

**Also the lack of such interim observations is a limitation of the study that needs to be acknowledged.**

We agree with the reviewer's comment and we have explained in Sect. 6.1 the influence of the Gloria storm to the results obtained and how they can be biased to accretive conditions (for example, we had to use high values of the onshore transport parameters in XBeach).

The lack of interim observations within the 6-month study period is a limitation of the study but it is also one of the main challenges that we tried to face in this study. Nevertheless, this issue has also been acknowledged in Sect. 6.1.

*Changes on Sect. 6.1 (page 28, lines 608 to 614).*

**Minor points**

**Contour plots like the ones in Figures 4 and 8 could look better with some smoothing or more interpolation in the colormap. I also think that these figures could be moved to the appendix.**

We agree with the reviewer suggestion so we have applied some smoothing into these figures and we have moved them to the Supplementary Information.

*Changes on Sect. D of the Supplementary Information (pages 10 and 12).*

**XBeach allows irregular or curvilinear grids which can deliver higher resolution close to the coast. Did the authors consider this? Is it a limitation that should be discussed?**

We only tried using a regular grid but we ensured that the grid size was fine enough to have accurate results. Indeed, using an irregular grid rather than the regular one could help to reduce the computational time. We have added a comment about this in Sect. 6.2.

*Changes on Sect. 6.2 (page 30, lines 659-660).*

**A typical issue where there are hard unerodable margins is the appearance of scouring/steps on the boundaries due to the local large sediment gradient. How do the authors deal with that in Q2Dmorpho? (I assume XBeach applies a lot of smoothing but it is already in the algorithm)**

We fully agree that this is potentially an issue in 2DH models but it was never a problem in Q2Dmorfo because it parameterises the effect of hydrodynamics into sediment transport. The alongshore transport is governed by the CERC formula so that it depends on the wave

angle relative to the local shore-normal (with certain smoothing). For example, starting with a rectilinear beach next to a headland  in the shadow zone, because the wave angle increases moving away from the headland, there is a sediment flux which is zero at the boundary and increases moving away from it. This causes sediment deficit at several mesh cells in the shadow zone, not only at the first one and results in a shoreline retreat there. Thus, the shoreline locally changes direction and tends to become normal to the local wave direction. Eventually a curvilinear shoreline shape with no net alongshore transport occurs (if the deep water waves remain constant) and potential steps or scouring channels are smoothed out.

However, at the beginning of this study we indeed had serious problems in XBeach related to this issue. When including solid groins in the lateral boundaries to simulate the rocky headlands, unrealistic scouring channels appeared. This is exactly the reason why we ended up using the pillars, as described in Sect. 3.1 and in the answer to a previous comment.

*Information on Sect. 3.1 (pages 10 and 11, lines 232 to 277).*

 **In Figure 6 I think that showing the bias (modelled-measured) rather than the actual values would be more of interest to the reader. Ideally the realizations could be replaced by a density scatter plot. (e.g. [https://es.mathworks.com/matlabcentral/fileexchange/95828-densityscatterchart](https://es.mathworks.com/matlabcentral/fileexchange/95828-densityscatterchart))**

We have added a figure showing the bias and modified the actual one applying opacity to the realization lines to represent the density of the coastline results.

*Changes on page 18, now Fig. 4.*

 **I couldn't find any information about the simulation times and the computational resources.**

As suggested before by the reviewer in Sect. 6.2 we have described the difference of computational times in the two models.

*Changes on Sect. 6.2 (page 30, lines 657 to 662).*

**Fig 9-10 Diverging colormaps would be more suitable for b (e.g. red-white- blue)**

We agree with the reviewer's comment and we have changed the figure into the suggested format.

*Changes on pages 20 and 21, now figure 6 and 7.*

**The paper is good and ready for publication even if two specific comments need to be addressed by the authors.**

**The introduction well precises the main goal of the study which is to test different wave forcing sources (in situ measurement, propagated buoy measurements and hindcasts) and compare the modelling results of two models on a 6 months period that is the good scale to compare these two models with different physic. The objective is to see what is the best source for modelling.**

**The studied site is correct except the needed of precision asked in the specific comments below. The physics of the numerical models used, calibrations, are well described. The bibliography is abundant.**

We thank the reviewer for the constructive and helpful comments on our manuscript. We have  implemented most of their suggestions, which improved the document. Below we respond to all comments. We underlined the sentences describing the specific changes implemented in the article and indicated the location of that changes in the version of the marked-up manuscript version of the paper.

**The results show not surprisingly that the necessity of in situ wave measurement at the entrance of the system is necessary even if propagated buoy measurement gives also satisfying results. Using hindcast for this specific embayed beach is not efficient due to problem of wave direction simulation. More the forcing data is close to reality, more results are good. This also demonstrates the need to have strong buoys networks to make run present model and future models for climatic response.**

We agree with the reviewer comment and have acknowledged this in the conclusion section.

*Changes on Conclusions (page 34, lines 758-759).*

**The results also shown the minor role of tide sea level in this Mediterranean low tide environment. This implies that wave gauge not exactly on the site can be used.**

**A discussion is made on the good results obtain for the X-Beach model for this large time scale and the essential role of the choose of the best parameters for the models, a comparison between the two models and the role of the forcing sources. This emphasis that having a good wave estimation of the wave direction is a key parameter for this embayed beach.**

**Conclusions are clear except the sentence addresses in specific comments.**

**Specific comments**

**- Even if the beach evolution is well explained with the rotation of the shoreline, the underwater shoreface needs to be more described in 1) the study area and 2) the bathymetric evolution on the base of the maps produced and the profiles shown. It seems that sand bars are present on the site, more information is needed and their morphological evolution done. The bathymetric profile in July shows a filling of the inner trough that confirm the post-Gloria recovery period studied. The offshore movement of the outer bar is more enigmatic without a vision of the allover form of the outer bar.**

We agree that a better description of the topobathymetry would be useful. There are no bars and troughs in the measured bathymetries (neither in the modelled ones) but we have included the information regarding the presence of terraces in the measured bathymetries in Sect. 2.2.

*Changes on Sect. 2.2 (page 5, lines 103-104).*

**-Lign 667-668: "On the other hand, the accuracy of the present simulations hardly depends on the sea level data source, even if tides are filtered, probably because they are small on many Mediterranean beaches." This seems to be contradictory with the main conclusions and the abstract that precises that "…did not depend on the sea level data sources"**

It is not completely clear to us what the reviewer is suggesting. We have understood that the reviewer finds it contradictory that in the abstract we wrote "did not depend" whilst in the conclusions we wrote "hardly depends". We have changed "did not" by "hardly" in the abstract to avoid confusion.

*Changes on Abstract (page 1, line 10).*